# Longitudinal variation in resilient psychosocial functioning is associated with ongoing cortical myelination and functional reorganization during adolescence

Meike D. Hettwer [1,2,3,4] ✉, Lena Dorfschmidt[5,6,7], Lara M. C. Puhlmann[4,8],
Linda M. Jacob[4], Casey Paquola[1], Richard A. I. Bethlehem [9], NSPN Consortium*,
Edward T. Bullmore[5], Simon B. Eickhoff[1,2,3] & Sofie L. Valk [1,2,3,4] ✉

Adolescence is a period of dynamic brain remodeling and susceptibility to psychiatric risk factors, mediated by the protracted consolidation of association cortices. Here, we investigated whether longitudinal variation in adolescents' resilience to psychosocial stressors during this vulnerable period is associated with ongoing myeloarchitectural maturation and consolidation of functional networks. We used repeated myelin-sensitive Magnetic Transfer (MT) and resting-state functional neuroimaging ($n = 141$), and captured adversity exposure by adverse life events, dysfunctional family settings, and socio-economic status at two timepoints, one to two years apart. Development toward more resilient psychosocial functioning was associated with increasing myelination in the anterolateral prefrontal cortex, which showed stabilized functional connectivity. Studying depth-specific intracortical MT profiles and the cortex-wide synchronization of myeloarchitectural maturation, we further observed wide-spread myeloarchitectural reconfiguration of association cortices paralleled by attenuated functional reorganization with increasingly resilient outcomes. Together, resilient/susceptible psychosocial functioning showed considerable intra-individual change associated with multi-modal cortical refinement processes at the local and system-level.

Adolescence is a period of pronounced brain remodeling that mediates biological and psychosocial maturation, but also heightened susceptibility to environmental adversity that may influence developmental trajectories[1,2]. The study of longitudinal trajectories in the presence of adversity exposure[3] and psychiatric symptoms[2,4,5] has thus been fundamental to advancing our understanding of inter- and intra-individual differences in psychiatric susceptibility. At the same time, there is a growing recognition that many individuals maintain good

[1]Institute of Neuroscience and Medicine, Brain & Behavior (INM-7), Research Centre Jülich, Jülich, Germany. [2]Max Planck School of Cognition, Leipzig, Germany. [3]Institute of Systems Neuroscience, Medical Faculty and University Hospital Düsseldorf, Heinrich Heine University Düsseldorf, Düsseldorf, Germany. [4]Max Planck Institute for Human Cognitive and Brain Sciences, Leipzig, Germany. [5]Department of Psychiatry, University of Cambridge, Cambridge, UK. [6]Lifespan Brain Institute, The Children's Hospital of Philadelphia and Penn Medicine, Philadelphia, PA, USA. [7]Department of Child and Adolescent Psychiatry and Behavioral Sciences, Children's Hospital of Philadelphia, Philadelphia, PA, USA. [8]Leibniz Institute for Resilience Research, Mainz, Germany. [9]Department of Psychology, University of Cambridge, Cambridge, UK. *A list of authors and their affiliations appears at the end of the paper. ✉e-mail: m.hettwer@fz-juelich.de; valk@cbs.mpg.de

mental well-being despite adversity, i.e., show resilient adaptation[6–8]. To comprehend bio-behavioral adaptation to an ever-changing environment, it has been vital to integrate neurodevelopmental assessments, complementary to inter-personal and physiological factors. Converging evidence from cross-sectional studies has highlighted brain regions involved in emotion regulation and stress reactivity in relation to adolescent susceptibility or resilience to environmental adversity. Specifically, resilient adaptation has been linked to larger prefrontal and hippocampal volumes, increased prefrontal regulation of amygdala activity, attenuated amygdala responses to adverse stimuli, and increased structural connectivity of the corpus callosum[9,10]. In the past decade, however, psychosocial conceptualizations have increasingly highlighted the dynamic nature of resilience[7,9,11–14]. Correspondingly, the ability to adapt to environmental adversity may show considerable intra-individual changes tied to plastic neurodevelopment. However, longitudinal studies exploring this notion, especially during periods of heightened susceptibility to psychopathology, remain scarce[1,9,15].

Insights into adolescent brain development have recently expanded from analyses of cortical size metrics (such as volume and thickness) to more fine-grained proxies of intra-cortical myelin maturation[5,16,17]. This line of research highlights the continuous myelination of intra- and inter-regional connections, enhancing circuit efficiency as a central feature of adolescent cortical maturation[18,19]. While myelination restricts structural plasticity by consolidating established connections, it has also been found to continuously modulate network dynamics to adapt to ever-changing environmental circumstances[19,20]. Rates of myelin maturation are heterochronous across the cortex and are particularly protracted in highly interconnected association cortices[16,21,22]. This protracted maturation, implying longer periods of developmental plasticity, likely reflects later refinement of functional networks associated with abstract cognitive functions, such as cognitive control. However, it also renders them more susceptible to environmental impact and psychopathological alterations[16,21,22]. Thus, the dual role of myelin in structural consolidation and dynamic functional adaptation makes the study of ongoing adolescent myelination a compelling focus to address the question of whether the maturation of behavioral capacities for psychosocial adaptation is tied to ongoing cortical consolidation.

Recent advances in in vivo imaging of cortical myelin have improved our understanding of myeloarchitectural maturation. One promising imaging contrast is magnetic transfer saturation (MT)[23], which is dominated by myelin-related molecules in the brain, as has been confirmed by several histological validation studies[24–26]. It has also been demonstrated to be sensitive to both developmental processes[5,16,17] and pathological alterations in myelin content[27]. Aiming at more nuanced insights into age-related changes in intracortical myeloarchitecture, several studies have sampled myelin-proxies across intra-cortical depths perpendicular to the cortical mantle, commonly referred to as "cortical profiling"[5,16]. Such depth-dependent profiling allows analysis of synchronized, large-scale patterns of cortical myeloarchitectural development by quantifying changes in interregional similarities (microstructural profile covariance; MPC). Previous work suggests that microstructural similarity predicts corticocortical connectivity[28,29]. Thus, studying changes in MPC with age yields valuable insights into system-level microstructural integration and differentiation, and its potential link to functional reorganization[30,31]. Association areas, in particular, represent a nexus of mixed intra-cortical profiles[32,33] that show marked and partly synchronized refinement well into early adulthood[16]. This microstructural refinement may be central to supporting the maturation of the intrinsic functional organization of the default and frontal parietal networks, supporting continued cognitive development of functions such as cognitive control and emotional flexibility[34,35]. In sum, leveraging multi-modal, system-level approaches is imperative to unravel

the complex role of cortical refinement in mental health and aligns with the broader understanding that maturational and psychopathological cortical alterations occur in a network-like fashion[36,37].

Together, previous research suggests that (1) understanding the development of psychosocial resilience requires complementary longitudinal studies, (2) the protracted consolidation of association cortices by myelination throughout adolescence likely confers increased susceptibility to adverse environmental influences, and (3) in vivo myelin mapping has facilitated multi-modal and multi-scale insights into cortical development. On this basis, the current study investigated whether intra-individual change in susceptibility and resilience to environmental adversity exposure is tied to differential rates of local and global myeloarchitectural consolidation, and accompanying functional maturation during adolescence and young adulthood (age range: 14–26 yrs). Environmental stressors included dysfunctional family environments, significant adverse life events, and low socioeconomic status, at two consecutive time points, one to two years apart. For each time point, we quantified a continuous resilient psychosocial functioning (RES$_{PSF}$) score, reflecting psychosocial functioning adjusted for individual stressor exposure. That is, RES$_{PSF}$ scores reflect residual variance in psychosocial distress that is not explained by the normative response to the stressor load an individual faced[7,38]. Thus, lower-than-expected distress reflects resilient adaptation, whereas higher-than-expected distress reflects greater susceptibility to environmental stressors[38]. We then investigated associations between intra-individual changes in resilient psychosocial functioning and brain maturation, specifically focusing on the role of ongoing myelination[18,21] and the impact on intrinsic function. We observed that longitudinal development towards more susceptible or resilient outcomes was associated with differential rates of prefrontal myelination, prefrontal functional network maturation, and cortex-wide myeloarchitectural reorganization of association cortices. Thus, extending cross-sectional studies suggesting increased susceptibility of association cortices to environmental impact[16,21,22], we conclude that adolescent cortical maturation of areas typically implicated in psychopathology is tied to dynamic intra-individual changes in psychosocial functioning relative to adversity.

## Results

### Resilient psychosocial functioning (Res$_{PSF}$) scores (Fig. 1)
We quantified continuous Res$_{PSF}$ scores by predicting psychosocial distress from measures of environmental adversity (Fig. 1A). Briefly, we derived a latent factor (Supplementary Table S1; chi$^2$ = 34293, $p < 0.001$), reflecting levels of psychosocial distress across domains of anxiety, depression, antisocial and compulsive-obsessive behavior, self-esteem, psychotic-like experiences, and mental well-being (similar to refs. 38,39). In a supervised random forest prediction, we then predicted psychosocial distress scores from adverse life events, childhood trauma, parenting style, family situation, and socioeconomic status (R$^2$ = 0.21, MAE = 15.15, correlation between true and predicted distress scores: $r$ = 0.46). The inverse deviations between true and predicted distress, i.e., the model residuals, were extracted to quantify resilient psychosocial functioning. Res$_{PSF}$ scores thus reflect a spectrum ranging from susceptible to resilient outcomes, i.e., the extent to which an individual shows higher or lower distress levels than expected given their stressor exposure (for similar approaches, see refs. 38,40–43). For parsimony, we will refer to this spectrum as resilient psychosocial functioning, which shall include the susceptible (negative) end of the spectrum. See Supplementary Table S2 for links to demographic data.

Different sub-samples were included across analyses in this study (Fig. 1A; see Supplementary Methods for details): The computation of distress score loadings ($n$ = 1533) and the prediction from adversity measures ($n$ = 712; subsample with additional NSPN U-change questionnaires) were conducted in independent samples to avoid leakage

effects. From the $n = 712$ sub-sample, we studied general patterns of MT maturation in $n = 199$ for whom longitudinal imaging data were available. We then linked longitudinal imaging patterns to change in resilient psychosocial functioning in $n = 141$ individuals who additionally completed all included questionnaires at repeated time points.

## Fundamental patterns of MT maturation

We started our investigations by evaluating fundamental myeloarchitectural patterns in the entire imaging sample ($n = 199$; 18.83 ± 2.84 y; 96 female), before addressing individual differences with respect to developing resilient psychosocial functioning. We visualized the group-averaged regional MT change between first and last imaging sessions (ΔMT) and observed a widespread intra-individual increase in MT (Fig. 1B; 1.26 ± 0.34 y apart). ΔMT was highest towards frontal and temporal poles with strongest inter-individual variability (std) in the

ventral prefrontal cortex. Systematic sampling of myelin-sensitive MT intensities along 10 equivolumetric surfaces perpendicular to the cortical mantle further revealed generally highest mean and interindividual variability in ΔMT in mid-to-deep layers.

## Intra-individual changes in resilient psychosocial functioning

To study longitudinal variation in resilient psychosocial functioning, we assessed the change (Δ) in $Res_{PSF}$ scores between the first and last measurement timepoint (on average 1.14 (SD 0.32) years apart) in $n = 141$ individuals for whom both repeated imaging and behavioral assessments were available (Fig. 1A). 57% of individuals showed a positive change in resilient psychosocial functioning with age (mean Δ = 2.40; SD = 16.35). We did not observe sex differences or age effects on changes in $Res_{PSF}$ scores in either this subsample or in the larger behavioral prediction sample in which $Res_{PSF}$ scores were calculated

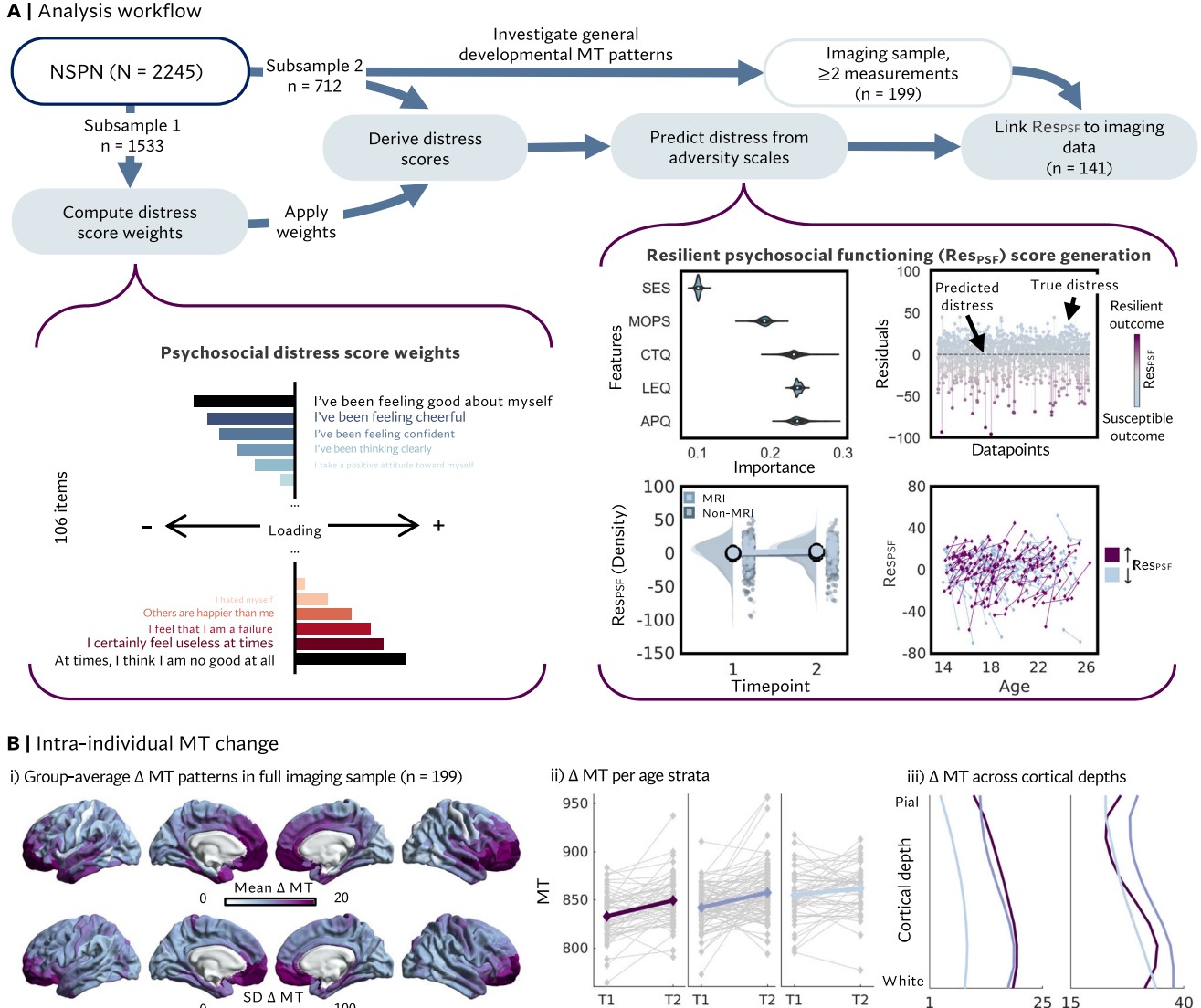

**Fig. 1 | Behavioral analysis workflow and group-average longitudinal change in myelin-sensitive Magnetic Transfer (MT). A** Based on the Neuroscience in Psychiatry Network (NSPN) cohort, resilient psychosocial functioning ($Res_{PSF}$) scores were computed for each subject at each available time point by predicting psychosocial distress (left) from adversity assessments (Alabama parenting questionnaire (APQ), Life events questionnaire (LEQ), Childhood trauma questionnaire (CTQ), Measure of Parenting style (MOPS), and socio-economic status (SES)). $Res_{PSF}$ scores were defined as the difference between observed and predicted distress, i.e., showing higher (i.e., more susceptible) or lower (i.e., more resilient) than expected psychosocial distress. Longitudinal changes in $Res_{PSF}$ are depicted in the bottom panel for participants that are part of the neuroimaging (MRI) or solely behavioral (non-MRI) analyses. **B** (i) Mean and standard deviation (SD) of intra-individual change in myelin-sensitive Magnetic Transfer (ΔMT) in the full imaging sample ($n = 199$). (ii) ΔMT averaged across the cortex and visualized for three age strata. (iii) Mean and SD of ΔMT across 10 intracortical depths and across the cortex. Line colors for (ii) and (iii) reflect age strata defined in the Middle panel.

**A | Intra-individual change in Res$_{PSF}$ and change in myelin-sensitive MT**

i) Association between Δ Res$_{PSF}$ and Δ MT

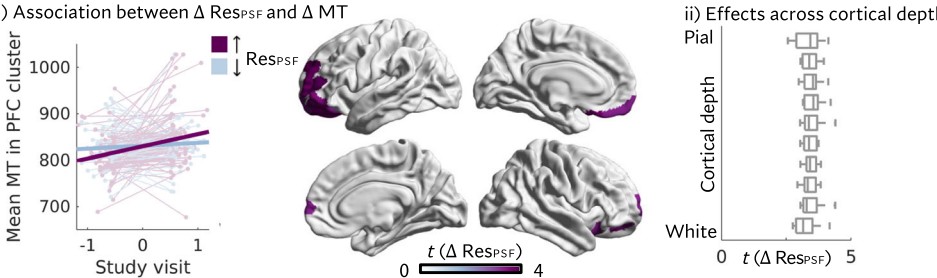

ii) Effects across cortical depths

**B | Cortical types**

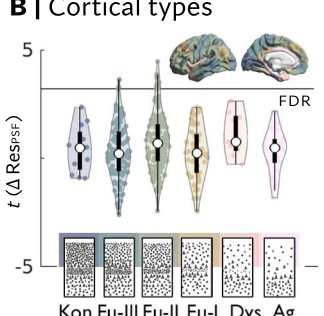

**C | Change in functional connectivity of the PFC cluster as a function of Δ Res$_{PSF}$**

i) Association between global prefrontal Δ FC and Δ Res$_{PSF}$

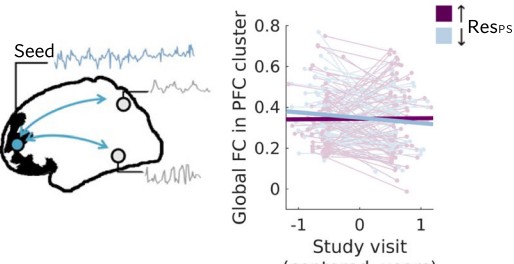

ii) Seed-based Δ FC and Δ Res$_{PSF}$

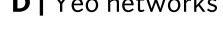
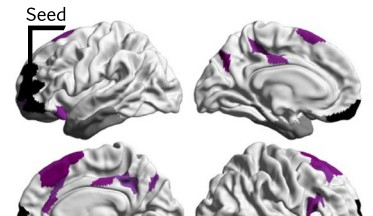

**D | Yeo networks**

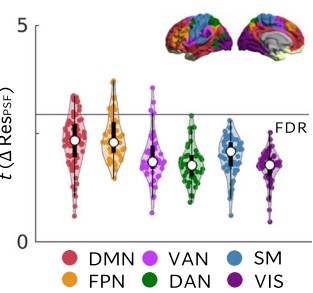

**Fig. 2 | Development of resilient psychosocial functioning is associated with changes in anterolateral prefrontal myelin-sensitive Magnetic Transfer (MT) and functional connectivity (FC). A** (i) A general linear model testing the association between change in resilient psychosocial functioning scores (Res$_{PSF}$) and change in MT (ΔMT) revealed that adolescents who showed increasingly resilient responses to psychosocial stressors with age showed a higher rate of myelin-sensitive MT change in the anterolateral prefrontal cortex (PFC; $p < 0.05$, FDR corrected, 10,000 permutations; two-sided test). (ii) This effect was on average homogeneous across cortical depths. Box plots for each intra-cortical surface include $t$-values derived for significant regions depicted in (i), where the box is defined by minima = 25% and maxima = 75%, lines depict medians, and whiskers are defined by values 1.5 times the interquartile range. Effects were predominantly located in eulaminate cortex II and III (**B**). Defining the cluster identified in (**A**) as a

seed (**C**), we further observed that prefrontal FC was more globally maintained (i) with increasing Res$_{PSF}$. Across the cortex (ii), this effect was most prominent in default mode (DMN) and frontoparietal (FPN) networks (D; $p < 0.05$, FDR corrected, 10,000 permutations; two-sided test). Gray masks in functional data reflect parcels that were excluded due to low signal-to-noise ratios. In (**B**) and (**D**), white dots reflect medians, violins depict vertical kernel density plots, the minima and maxima of black boxes are defined by 25% and 75% quartiles. All results depicted here are based on $n = 141$ individuals. Note that line plots in Ai) and Ci) are colored with respect to increasing vs. decreasing Res$_{PSF}$ for visualization, but analyses were performed on continuous Res$_{PSF}$ scores. Kon = Konicortex, Eu-I-III = Eulaminate I-III, Dys = Dysgranular, Ag = Agranular; VAN = Ventral attention network, DAN = Dorsal attention network, SM = Sensorimotor, VIS = Visual.

($n = 455$ out of $n = 712$ individuals with at least two measurement time points) (Supplementary Table S2). Longitudinal changes in resilient psychosocial functioning were not related to changes in stressor exposure (Supplementary Fig. S1) and showed comparable distributions in individuals included in the imaging analyses and individuals included in the behavioral analyses only (Supplementary Fig. S2).

**Intra-individual variation in resilient psychosocial functioning and myelin-sensitive MT (Fig. 2)**

Once longitudinal MT patterns and Res$_{PSF}$ scores were determined, we aimed to elucidate the association between ongoing myelination during adolescence and changes in resilient psychosocial functioning ($n = 141$). We observed a positive association between developing toward more resilient functioning (i.e., an intra-individual increase in Res$_{PSF}$ scores) and ΔMT in the predominantly left-lateralized anterolateral prefrontal cortex (PFC; $t_{max}(134) = 4.51$; $\beta_{standardized} = 0.36$; $CI_{standardized\ \beta} = [0.20, 0.52]$ (medium effect); $p_{10,000\ permutations\ \&\ FDR} < 0.05$; Fig. 2A). This effect was robust to several analytical choices and sub-sampling (see Supplementary Fig. S3). Given that myelination rates are not homogeneous across cortical depths (see Fig. 1B iii), we further tested for intracortical differentiability of the observed effect within the prefrontal cluster. The positive association between longitudinal variation in resilient psychosocial functioning and ΔMT was homogeneous across 10 intra-cortical sampling depths (Fig. 2A ii).

We next assessed whether the effects of longitudinal variation in resilient psychosocial functioning were concentrated in regions characterized by a specific cytoarchitecture, and associated duration of developmental plasticity, using cortical types. The five cortical types included agranular, dysgranular, eulaminate I, II and III, and koniocortex and have been proposed to represent a hierarchy of cortical architectonics, ranging from highly differentiated and myelinated koniocortex to less differentiated and more plastic agranular cortex[28,44]. We stratified the unthresholded t-map according to this prior categorization of cortical types and identified which cortical types overlapped with the significant prefrontal cluster. We observed that parcels showing a significant effect of changes in Res$_{PSF}$ scores were located in eulaminate cortex II & III (Fig. 2B), which contain regions of comparatively high cytoarchitectural complexity and layer differentiation. Overall, the cortical topology of the unthresholded t-map followed a general posterior-to-anterior pattern, aligning with a cortex-wide axis of MT development (Supplementary Fig. S4)

Probing whether observed longitudinal effects might be related to cross-sectional differences at baseline, we observed no cross-sectional association between Res$_{PSF}$ scores and MT. One medial frontal gyrus parcel showed lower baseline MT in individuals with lower Res$_{PSF}$ scores at baseline compared to follow-up (see Supplementary Fig. S5).

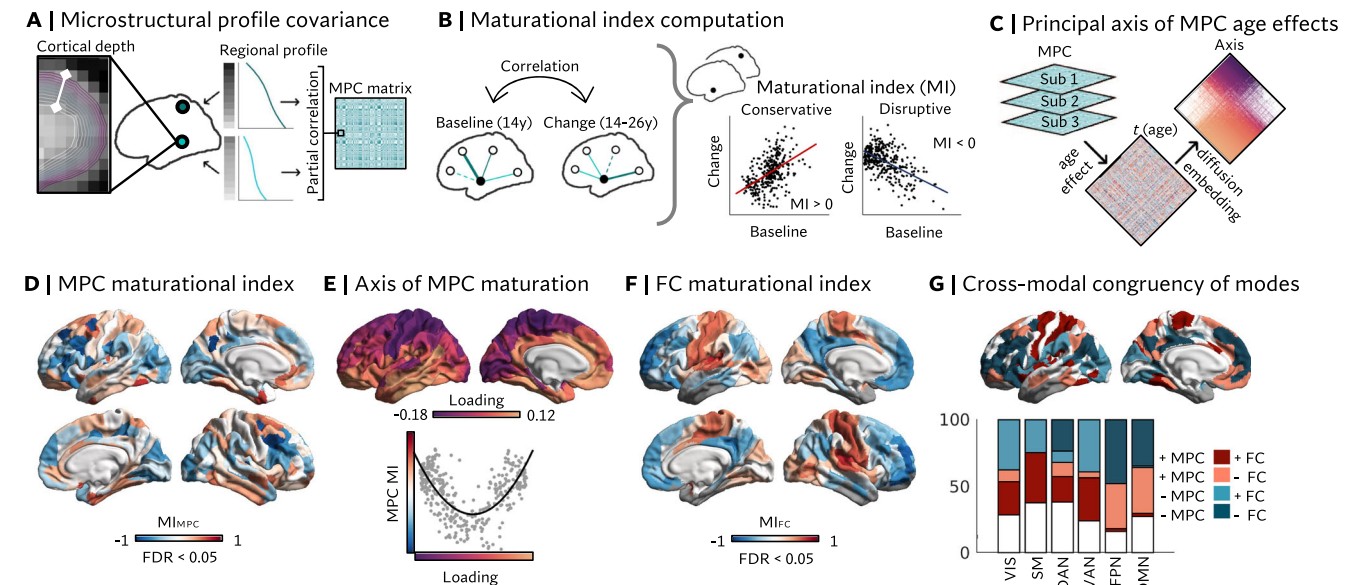

**Fig. 3 | Systems-level cortical maturation in the full imaging sample (n = 199), capturing multi-modal reorganization in association cortices. A–C** Depict analytical approaches. **A** MT intensities were sampled along 10 equi-volumetric surfaces between gray matter (pial) and gray matter /white matter boundaries to derive microstructural profiles and a microstructural profile covariance (MPC) matrix. **B** The maturational index (MI) captures correlations between baseline (i.e., a pattern predicted for age 14 by a mixed effects general linear model) and change patterns (i.e., the age effect estimated by that model) in a region's network. **C** A low-dimensional axis of MPC age effects was derived by applying diffusion map embedding to order regions according to their similarity in synchronized micro-structural differentiation with age. **D–G** depict derived maturational patterns. **D** MI of MPC (MI$_{MPC}$; pFDR <0.05), showing conservative development in ventral temporal and dorsal regions, and disruptive reorganization in fronto-parietal association cortex. **E** Relationship between the MI$_{MPC}$ and the principal axis of MPC maturation. The non-linear relationship indicates that conservatively developing ventral and dorsal regions follow maximally different developmental patterns. **F** MI of functional connectivity (MI$_{FC}$; pFDR <0.05), showing conservative development in sensorimotor cortex and disruptive reorganization in heteromodal association cortex. **G** Maturational categories: Overlaps between MI$_{MPC}$ and MI$_{FC}$ per Yeo network. + = conservative, - = disruptive. VIS Visual, SM Sensorimotor, DAN Dorsal attention network, VAN Ventral attention network, FPN Frontoparietal network, DMN Default mode network. Gray masks in functional connectivity (FC) data reflect parcels that were excluded due to low signal-to-noise ratios.

## Intra-individual change in intrinsic functional connectivity of the anterolateral prefrontal cortex

Having established a positive association between myelin-sensitive MT increase and intra-individual increases in resilient psychosocial functioning, we next investigated concordant changes in the prefrontal cluster's intrinsic functional connectivity (Fig. 2C). To this end, we defined the identified prefrontal parcels exhibiting a significant effect in MT analyses as a seed and assessed both global (i.e., degree centrality) and network-level effects of changes in Res$_{PSF}$ scores on intrinsic functional connectivity (ΔFC). Globally, we observed more maintained levels of cortex-wide functional connectivity with increasingly resilient outcomes, whereas increasingly susceptible outcomes were associated with a segregation (small global effect: t(134) = 2.45; β$_{standardized}$ = 0.21; CI$_{standardized\ β}$ = [0.04, 0.37], $p = 0.02$). Studying the cortex-wide pattern of associations between changes in Res$_{PSF}$ scores and prefrontal ΔFC revealed that effects were concentrated in regions of the default mode, frontoparietal and ventral attention networks (medium regional effect: t$_{max}$(134) = 3.78; β$_{standardized}$ = 0.31; CI$_{standardized\ β}$ = [0.14, 0.46], p$_{10,000\ permutations\ \&\ FDR}$ < 0.05; Fig. 2C, D) and were driven by sub-regions of the PFC cluster that are part of the default mode network (see Supplementary Fig. S6).

## System-level cortical maturation (Fig. 3)

Thus far, our analyses suggest a role of local prefrontal myeloarchitectural and inter-regional functional network maturation for developmental changes in susceptible/resilient psychosocial functioning. This suggests that local microstructural alterations may also reflect system-level cortical refinement. Therefore, we next aimed to study system-level myeloarchitectural and parallel functional reorganization. To this end, we computed a microstructural profile covariance (MPC) network reflecting interregional similarities of myeloarchitectural profiles. The MPC matrix was generated by first probing MT intensities at ten equally spaced intra-cortical depth coordinates, yielding cortical depth profiles of regional MT from the pial surface to the white matter boundary of each cortical area. We then calculated the pairwise Pearson correlation between regional profiles while controlling for average MT intensity to derive the MPC matrix (Fig. 3A). This allowed us to examine the topology of synchronized effects of age on depth-specific changes in approximated myelin content, which are reflected in changes in regional intra-cortical profiles and thus their inter-regional similarity. Next, we computed a maturational index (MI$_{MPC}$; Fig. 3B), which captures the age-related change of all MPC edges of a node as a function of their respective baseline patterns (estimated for age 14[45]). The MI$_{MPC}$ revealed a topologically heterogeneous pattern of reorganization (p < 0.05, FDR) with strongest reorganization in frontoparietal association cortices (Fig. 3D). This pattern was robust to sub-sampling (Supplementary Fig. S7). The MI$_{MPC}$ was spatially aligned with a previously established cortical axis of age-related MPC change (Fig. 3C). Regions closer on this axis exhibit more similar patterns of age-related change in MPC, whereas distant regions undergo dissimilar development[16]. The axis captures a differentiation of frontoparietal association cortices that we found to exhibit disruptive reorganization, to resemble either idiotypic sensory or paralimbic/temporal cortex maturational patterns (Fig. 3E). At the same time, we observed a U-shaped association between the MI$_{MPC}$ and the main axis of age effects, where regions at the extremes of the axis exhibited a positive MI$_{MPC}$, i.e., a positive correlation between baseline and change patterns. A positive MI$_{MPC}$ reflects an integration of regions that showed higher myeloarchitectural similarity at baseline and/or a differentiation of regions that were already dissimilar at baseline. This strengthening of existing patterns has been termed 'conservative' development[45]. Conversely,

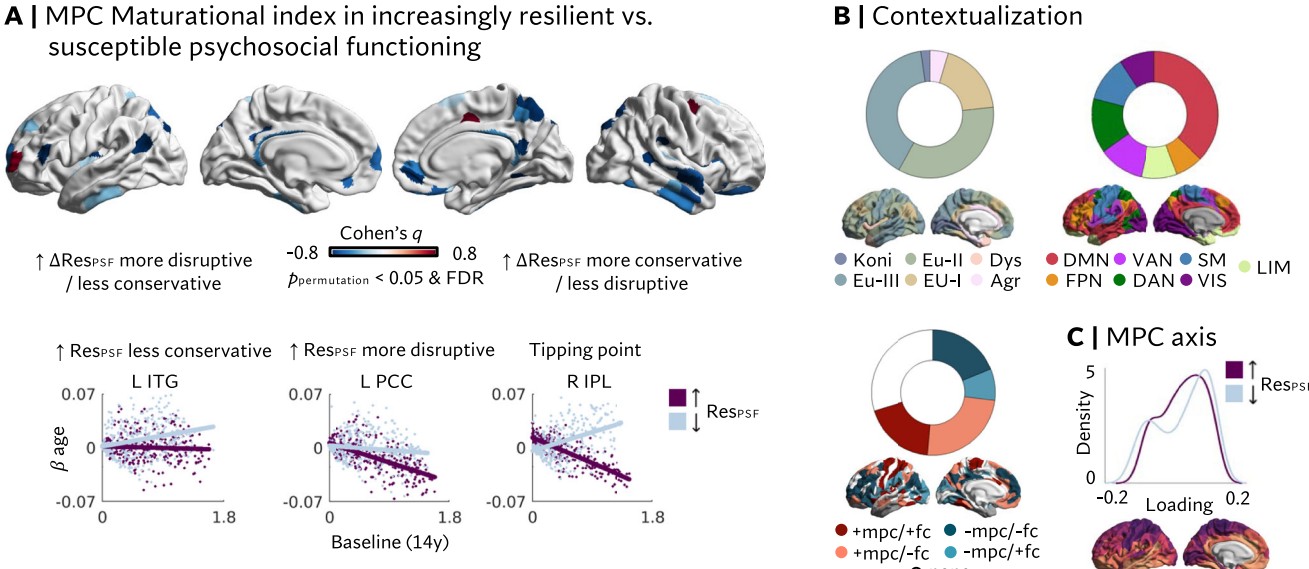

**A | MPC Maturational index in increasingly resilient vs. susceptible psychosocial functioning**

↑ΔResPSF more disruptive / less conservative

Cohen's q
-0.8 ⬤⬤⬤ 0.8
$p_{permutation} < 0.05$ & FDR

↑ΔResPSF more conservative / less disruptive

↑ ResPSF less conservative — L ITG
↑ ResPSF more disruptive — L PCC
Tipping point — R IPL

β age: 0.07, 0, -0.07
Baseline (14y): 0, 1.8

⬛ ↑ ResPSF
⬛ ↓ ResPSF

**B | Contextualization**

⬤ Koni  ⬤ Eu-II  ⬤ Dys     ⬤ DMN  ⬤ VAN  ⬤ SM     ⬤ LIM
⬤ Eu-III ⬤ EU-I ⬤ Agr     ⬤ FPN  ⬤ DAN  ⬤ VIS

⬤ +mpc/+fc  ⬤ -mpc/-fc
⬤ +mpc/-fc  ⬤ -mpc/+fc
○ none

**C | MPC axis**

Density: 5, 0
Loading: -0.2, 0.2
⬛ ↑ ResPSF
⬛ ↓ ResPSF

**Fig. 4 | More disruptive reorganization of microstructural profile covariance (MPC) networks with increasingly resilient mental health outcomes. A** Top; Cohen's q for group differences in the microstructural Maturational Index (MI$_{MPC}$; FDR $p < 0.05$ & 10,000 permutations, $p < 0.05$) indicating a widespread negative shift (i.e., more disruptive reorganization or less conservative MPC development) in the group of individuals who developed towards more resilient outcomes (↑ΔRes$_{PSF}$). Group differences were computed via z-tests and significance was assessed based on both $p < 0.05$ FDR and non-parametric permutation testing (10,000 permutations; $p < 0.05$). Tests were two-sided. Bottom; A negative shift for the ↑Δ Res$_{PSF}$ group can reflect three main scenarios: Less conservative development (left), more disruptive development (middle), or disruptive development in a region that exhibits neither disruptive nor conservative development in the full sample (right; tipping point). **B** Contextualization of regions with a significant group difference (in any direction), revealing that differences in MI$_{MPC}$ are more frequently located in Eulaminate cortex II & III, default mode areas, and regions exhibiting decoupled microstructural compared to functional development. **C** Density plot of axis loadings per group, reflecting a more compressed axis in the group of individuals who developed towards more resilient outcomes. For scatter and density plots, the ↑Δ RES$_{PSF}$ group is depicted in purple and the ↓ RES$_{PSF}$ group in blue. Statistical tests were two-sided. Kon Konicortex, Eu-I-III Eulaminate I-III, Dys Dysgranular, Ag Agranular, DMN Default mode network, FPN Frontoparietal network, VAN Ventral attention network, DAN Dorsal attention network, SM Sensorimotor, VIS Visual, LIM Limbic.

towards the center of the axis, we observed a negative MI$_{MPC}$, indicating a reorganization of MPC embedding. Here, regions that were more dissimilar at baseline became more integrated with each other and/or regions with higher myeloarchitectural similarity became more differentiated. This is termed 'disruptive' reorganization (which does not imply pathological disruption, but rather a disruption of baseline patterns during development).

Given the complex relationship between system-level structural and functional development[34,46], we studied the convergence of myeloarchitectural and functional maturational modes. The MI based on functional connectivity (MI$_{FC}$) is characterized by a clear differentiation of 'conservative' development in unimodal regions and 'disruptive' reorganization in heteromodal association areas (Fig. 3F[45]). We found parallel conservative development as well as MPC-reorganization co-occurring with conservative FC development in regions involved in sensory- and attention-related processes (visual, sensorimotor, and ventral attention network). Regions of the default mode and frontoparietal networks showed both cross-modal reorganization and structure-function divergence, with MPC showing conservative but FC disruptive developmental patterns (Fig. 3G). Together, this shows that microstructural and intrinsic functional organization show both convergent and divergent system-level maturational patterns.

**System-level maturation and intra-individual change in resilient psychosocial functioning (Fig. 4)**

Last, we investigated whether longitudinal changes in resilient psychosocial functioning are associated with different degrees of system-level reorganization between the ages of 14 and 26. Because both the MI and the main axis of age-related MPC change are derived from group-level statistics, it was required to form groups for these analyses

by dichotomizing behavioral changes reflecting increasingly resilient (+ΔRes$_{PSF}$) vs. increasingly susceptible (−ΔRes$_{PSF}$) outcomes. We observed significant group differences between +ΔRes$_{PSF}$ and -ΔRes$_{PSF}$ individuals in the MI$_{MPC}$ in 43 predominantly heteromodal regions (Fig. 4A; $p < 0.05$ FDR & 10,000 permutations). 93% of these regions showed a negative shift in MI$_{MPC}$ in individuals developing towards more resilient outcomes, mostly reflecting less conservative development (58%; Fig. 4B; Supplementary Table S3), but also more disruptive reorganization (19%). A further 16% of these reflected regions that generally showed no significant association between baseline and age-related change patterns in the full imaging sample, that is, in regions that followed neither conservative nor disruptive developmental patterns. The observations were robust to alternative modeling approaches (Supplementary Fig. S8).

Contextualizing these findings with existing atlases of cortical types[28] and intrinsic functional networks[47] revealed that group differences were concentrated in Eulaminate-II (35%) and -III (40%) cortex, defined anatomically, and the default mode network (DMN; 37%), defined functionally (Fig. 4B). In a complementary approach, we investigated MPC maturation along a low-dimensional cortical axis. We observed a more pronounced bimodal distribution of loadings along the principal axis in the group of individuals who developed toward higher susceptibility. That is, regions were situated toward the differentiated apices rather than the middle of the axis, reflecting increasing microstructural similarity to the axis' anchors. Conversely, the axis was slightly compressed in the group of individuals who developed toward more resilient outcomes. Here, more parcels loaded on the middle of the axis reflecting less synchronization with the anchor profiles (Fig. 4C).

Throughout this study, functional connectivity data were used to contextualize the myeloarchitectural results. Considering the

topology of cross-modal congruency of MIs (Fig. 3G), we observed that group differences in $MI_{MPC}$ were largely located in regions where MPC and FC did not follow convergent MI patterns (see Fig. 4B). That is, group differences were most frequently located in regions exhibiting conservative MPC but decoupled, disruptive FC development (35%). The $MI_{FC}$ itself showed subtle group differences in eight confined, primarily prefrontal regions (Supplementary Fig. S9). Five of these regions showed less disruptive, three exhibited less conservative development in the group of individuals who became more resilient with age.

Together, this suggests that both myeloarchitecture and functional connectivity show marked maturational reorganization tied to resilient/susceptible changes in psychosocial functioning, establishing the brain as a key feature of adaptive development.

## Discussion

In the current study, we report that intra-individual variation in adolescent psychosocial functioning relative to environmental adversity is tied to ongoing myeloarchitectural and functional maturation of association cortices. Here, resilient and susceptible outcomes were operationalized as comparatively lower or higher levels of psychosocial distress in the face of psychosocial stressors, resulting in a continuous score that adjusts for variations in stressor exposure at different time points. We used a dimensional approach aligning with previous studies that underscore psychosocial adversity as a pivotal transdiagnostic risk factor that is more predictive of overall psychopathology than discrete symptom domains[48,49]. In addition, our longitudinal design puts emphasis on the dynamic nature of resilient functioning, combined with myeloarchitectural brain phenotypes that account for regional variations along cortical depth. Thus, we provide nuanced and multimodal evidence that protracted maturation of association cortices is associated with changing abilities to adapt to psychosocial stressors during adolescence.

### Enhanced anterolateral prefrontal myelination links to increasingly resilient psychosocial functioning

Investigating longitudinal changes in myelin revealed a positive association between ΔMT in anterolateral and orbitofrontal cortex and changes in resilient psychosocial functioning at medium effect size. This finding is consistent with previous cross-sectional reports suggesting a particular susceptibility of the ventral prefrontal cortex to environmental adversity[3,22,50] and central role in resilient adaptation[10,48]. Here, we extend previous findings on cortical volumes and functional connectivity toward longitudinal myelin plasticity. A beneficial effect of enhanced prefrontal myelination may directly be linked to the optimization of adaptive cognitive strategies facilitating successful navigation in an ever-changing environment[51]. That is, ongoing plasticity of myelination fosters circuit modification and synchronization through a multitude of parallel mechanisms[20]. These may include regulatory influences on axon conductance to optimize the synchronization of spike arrivals[52,53], neuronal metabolism and excitability[54,55], and structural plasticity[14,56]. In the prefrontal cortex, the optimization of circuit efficiency is closely linked to the maturation of cognitive functions such as executive functions, including emotion regulation, and enhanced social and cognitive flexibility required for adaptation[57,58]. Thus, prefrontal maturation may directly facilitate resilient psychosocial functioning by fostering cognitive strategies such as cognitive reappraisal, self-awareness about potential maladaptive cognitive biases, or decision making/ problem solving to evaluate the impact of adverse experiences and, for example, seek social support. Thus, the acquisition of beneficial cognitive strategies may therefore mediate the positive association between prefrontal maturation and resilient psychosocial functioning. Conversely, attenuated prefrontal myelination and impaired executive control have been linked to transdiagnostic mental health impairments[59–61].

Schizophrenia rat models further suggest links between interneuron hypomyelination and cognitive inflexibility[62]. It is noteworthy that adolescents exhibiting increasingly susceptible outcomes in the present study did not surpass clinical thresholds. However, current findings indicate that cross-sectional associations between susceptibility to psychopathological spectra and prefrontal myeloarchitectural development described in patient and animal data can already be observed at the level of intra-individual variation in susceptibility. Lastly, in addition to potential cognitive effects, psychosocial adversity is likely to elicit a physiological stress response activating the Hypothalamus-Pituitary-Adrenal axis. The ventral PFC is involved in and can recursively be affected by physiological stress responses through glucocorticoid-induced structural remodeling[63,64]. While the current data do not allow to test protective effects at the molecular level, it is possible that increased consolidation of connections through myelination may enhance physiological resistance to adverse stressor-induced PFC remodeling.

We probed whether varying levels of myeloarchitectural consolidation coincided with differences in functional network maturation of the identified anterolateral prefrontal region, indirectly suggesting a potential resistance to stressor-induced remodeling. Indeed, we observed greater stability of PFC connectivity among individuals who developed toward more resilient psychosocial functioning. Conversely, increasingly susceptible mental health outcomes were associated not only with a reduced rate of prefrontal myelination, but also with a segregation of prefrontal connectivity within abstract cognitive networks (small global effect, medium regional effects). In normative development, most prefrontal sub-regions show decreases in global network embedding during childhood followed by a plateau in early adulthood[65–67]. This pattern reflects a combination of increasing integration within networks of which they are part, such as the DMN and FPN, but a segregation from other networks such as the dorsal attention network. Here, the segregation of the anterolateral PFC region both globally and within the DMN and FPN in individuals manifesting increasingly susceptible outcomes suggests a closer tie to patterns reminiscent of earlier developmental stages in both prefrontal connectivity and mean myelin-sensitive MT. Together, increased longitudinal myelination of anterolateral prefrontal regions may link to facilitated adaptation to adversity by optimizing the efficiency of prefrontal cognitive circuits relevant to flexible adaptation and resistance to adverse remodeling.

### Synchronized reorganization of regions higher up the cytoarchitectonic and functional hierarchies are implicated in developing resilient psychosocial functioning

Brain alterations linked to both maturational and psychopathological cortical alterations have been proposed to occur in a network-like fashion rather than in isolation[36,37], underlining the importance of considering the embedding of local changes in a globally changing system. Therefore, we studied the cortical topology of synchronized maturation of intracortical profiles, to then probe whether areas exhibiting most pronounced reorganization during adolescence are more strongly implicated in the development of resilient psychosocial functioning. We described a maturational index reflecting age-related change as a function of baseline patterns and observed the most profound reorganization in frontoparietal association cortices. The $MI_{MPC}$ pattern aligned with a previously established principal axis of age-related MPC change that suggests fronto-parietal association cortices to differentiate most profoundly in synchronization with either dorsal/unimodal or ventral/paralimbic anchors[16]. Importantly, the microstructural maturational topology differs to some extent from the maturational topology that has been previously described for functional networks[45]. In particular, cognitive networks such as the FPN and DMN exhibited not only congruent disruptive development, but also a structure-function de-coupling marked by conservative MPC

but disruptive FC development. The partially independent remodeling of functional and structural connectivity has been suggested to shape functional specialization in transmodal association cortex critical for executive functions[34]. At the same time, it underscores the importance of multimodal studies in understanding the consequences of system-level maturation.

We observed associations between MI and changes in resilient psychosocial functioning in microstructural, and regionally confined effects in intrinsic functional data. Compared to individuals developing toward higher stressor susceptibility, adolescents developing toward more resilient outcomes exhibited a negative-shift in the microstructural $MI_{MPC}$ across heteromodal association cortices. This negative-shift reflected both less conservative and more disruptive development. The observed reduction in conservative development was further highlighted by a compressed axis of MPC change, with fewer regions loading toward the apices/anchors of the axis. In contrast to the $\Delta$ $Res_{PSF}$ effects observed in $MI_{MPC}$, $MI_{FC}$ effects were more locally concentrated in the bilateral prefrontal cortex and presented a positive rather than negative shift in $MI_{FC}$ in individuals who became more resilient with age. That is, the $+\Delta$ $Res_{PSF}$ group exhibited less disruptive development in the PFC. This finding is complementary to microstructural maturational patterns associated with longitudinal change in resilient functioning and converges with the observation that increasingly resilient outcomes were associated with more maintained prefrontal functional connectivity (see Fig. 2C). Previous work suggests that regions that are most developmentally active during adolescence, primarily association cortices, are most strongly implicated in mental health[1]. While we generally observed effects in association cortices, this was not exclusive to regions exhibiting disruptive reorganization. In particular for the microstructural $MI_{MPC}$, effects associated with changes in resilient psychosocial functioning were marked in the temporal cortex, which in turn exhibited largely conservative development. Our results thus indicate that longitudinal variation in resilient psychosocial functioning is linked to altered degrees of myeloarchitectural reorganization, but this was independent from whether a region generally developed conservatively or disruptively.

Across the analytical scales and imaging modalities included in this study, regions implicated in longitudinal change in resilient psychosocial functioning were characterized by their high position along cortical hierarchies of cytoarchitectonic complexity and functional network abstraction. That is, findings emerged predominantly in the cytoarchitectonically complex eulaminate-II and -III cortices, anatomically, and the DMN, functionally. Previous research suggests that structural differentiation of the DMN from networks involved in sensory-perceptual processing[46] facilitates the maturation of cognitive functions requiring abstraction from the immediate environment[57,58]. At the same time, DMN structure and functional connectivity are often implicated in psychiatric symptom domains, exposure to environmental adversity such as low socio-economic-status, but also to protective environmental factors such as positive parenting[68–70]. A prominent explanation for the recurrent role of the DMN is its involvement in the generation of conceptual mental models of the self in the environment[71–74]. Such self-in-context models include self-referential processing, emotional reappraisal, assigning meaning to external events and interpreting their causes in reference to one's own narrative. Maladaptive internal models and inaccurate attributions of causality fostering negative interpretations of experiences have been considered transdiagnostic risk factors for mental health impairments[69,71]. Similarly, resilient or susceptible psychosocial trajectories may be tied to continuously evolving self-in-context representations. Here, ongoing refinement of the DMN may facilitate and stabilize beneficial self-referential mental narratives that influence adaptive strategies in the face of environmental stressors.

Lastly, investigating both an average myelin proxy (i.e., regional MT) and a more nuanced approximation of intra-cortical profiles highlighted different facets of the associations between cortical maturation and change in resilient psychosocial functioning. Nevertheless, both perspectives support a beneficial role of myelin plasticity, as reflected in higher overall rates of (anterolateral prefrontal) myelin growth and, when considering depth-specific measures, higher levels of microstructural reorganization. Across present analyses, individuals who developed toward more susceptible outcomes showed maturational profiles more closely tied to patterns associated with earlier stages of adolescence. Conversely, microstructural maturation in individuals with increasingly resilient outcomes appeared less constrained by existing patterns, potentially highlighting the need for adaptive alterations to enhance selected cognitive circuits. Overall, it is likely that not only is the more the better, but that parallel refinement processes occur at multiple scales. Thus, the current findings demonstrate that adolescents exhibit marked intra-individual variability in resilient psychosocial functioning relative to environmental adversity that is reflected in both local and system-level brain maturation profiles.

## Limitations, further considerations, & open questions

Our study takes a dimensional approach to environmental adversity exposure and psychosocial functioning. While differences exist in the brain correlates of both adversity type[50] and symptom domains[4], we believe a dimensional approach enhances ecological validity as included forms of adversity are highly clustered together in the general population[75], have been shown to be associated with overlapping brain structural correlates (dice coefficients up to 0.54[3,50]), and are transdiagnostic predictors of overall psychopathology[48,49]. We acknowledge that our analyses of resilient psychosocial functioning are limited to the inclusion of only two timepoints. While this allows us to study longitudinal variation, we note that more timepoints per participant would be required to estimate trajectories with higher reliability[76]. Future studies utilizing e.g., later release waves of the Adolescent Brain Cognitive Development (ABCD) cohort could track longitudinal trajectories over a longer period of time and further assess the question whether enhanced myelin maturation is also a predictor for adult mental health[49]. Next, current evidence suggests that individual differences in myeloarchitectural maturation may be a potential neurobiological resilience factor, influencing adolescent adaptation to environmental risk factors. The exact mechanisms underlying a potential protective effect cannot be clearly elucidated in the current study due to its correlational nature but may involve increased structural stability to stress-induced remodeling and enhanced cognitive maturation. It is likely that resilient psychosocial development is closely coupled with the attainment of cognitive strategies that facilitate resilient outcomes[77]. At the same time, environmental resilience factors such as social support facilitate resilient outcomes[38,49,78], and may in part exert their protective effect through an impact on brain maturational trajectories. Resilient outcomes are assumed to rely on a multi-modal and multi-faceted construct, acknowledging the environments we live in, but also other psychological variables beyond clinical symptomatology (such as positive affect, life satisfaction, personality traits). We cannot clearly disentangle the interaction of different intrinsic and extrinsic influences contributing to an individual's psychological well-being beyond resilience. While we estimated resilience/susceptibility by adjusting psychosocial well-being for adversity exposure - yielding a residualized psychosocial functioning score - we cannot rule out that derived resilience scores also reflect the influence of other genetic and environmental factors, as well as noise or measurement error to a certain degree (see Supplementary Information). For example, potential self-report/retrospectivity biases inherent to measures of well-being and adversity exposure[79] would persist in resilience scores, but are not caused by the residual approach to providing resilience scores. Our longitudinal approach allowed us to limit confounding effects of individual differences in genetic predispositions or environmental circumstances (such as family composition

or neighborhood) that are likely to contribute less variability to within-subject repeated measures than to between-subject cross-sectional data. Overall, our model (explaining 21% of the variance) controls for exposure to a similar extent as common resilience models (explaining 21-28% of the variance; see e.g., refs. [38,80,81]). We also aimed to increase the robustness and generalizability of the model by implementing a nested cross-validation and a random forest regression robust to non-linear and non-parametric distributions of questionnaire data. Next, we observed a more skewed distribution of SES in the imaging sub-sample (Supplementary Fig. S2), suggesting a potential over-sampling of individuals from a comparatively more affluent background. Brain-behavior associations described here may thus be limited by the reduced variance in SES. This common issue in developmental neuroimaging research[82] demands the study of resilience factors identified by the current study in specific sub-groups, such as cohorts facing specific economic difficulties, that were under-represented in the current sample. However, we also note that distributions in well-being, as well as other risk exposure assessments, did not differ between imaging and non-imaging sub-samples, implying that the over-representation of higher SES participants in the imaging sub-sample was not associated with a commensurate shift in the distributions of risk exposure measures that were weighted more strongly in the prediction of $Res_{PSF}$. We further acknowledge that this sample is of respectable but not massive size. The current sample size resulted from the inclusion of a sample for which adolescent, longitudinal, and multi-modal imaging including a myelin proxy, as well as in-depth phenotypic characterization, were available—rather than an a-priori power analysis. Overall, we found the reported results to be robust to model parameter manipulation and sub-sampling, and well in line with the existing literature, highlighting the central role of the PFC in stress adaptivity and vulnerability[9,83], as well as the role of association cortex maturation in psychiatric vulnerability[1,21,22]. Moreover, individual differences in myelination rates were observed in regions that generally show the highest rates of myelination during adolescence (Fig. 1B). This supports a link to the protracted critical period of plasticity in the prefrontal cortex, extending well into early adulthood and associated with increased susceptibility to environmental risk factors[22]. We further observed convergence across imaging modalities, for example, effects on the maturation of prefrontal functional network that substantively corroborate the observed differences in rates of prefrontal myelination. However, recent reviews of the replicability of neuroimaging studies in the context of insufficient sample sizes[84] have cautioned against inflated effect sizes. Therefore, we emphasize that the current results should be interpreted with caution, pending replication in future studies with independent and larger samples, more representative of diverse cultural backgrounds and including individuals typically excluded from healthy samples, such as individuals with neurodevelopmental disorders, to assess the broader generalizability of neurobiological resilience factors identified here. Last, complementary to longitudinal approaches aimed at making prospective predictions of future mental health outcomes, our current work argues for tracking ongoing developmental trajectories to better understand of intra-individual variability in susceptibility to environmental risk factors at different time points in development.

To conclude, the transition to adulthood is considered a particularly susceptible period for the emergence of mental health symptoms. Consistent with prior research suggesting a central role of the protracted development of association cortices in susceptibility to psychiatric risk factors[21], the current work suggests that intra-individual changes in psychosocial responses to environmental stressors are associated with the degree of myeloarchitectural plasticity and cortex-wide reorganization. The dynamic nature of myelin suggests a potential benefit of interventions that target aberrant trajectories in at-risk youth. These may include increased exposure to environmental resilience factors such as a supportive social network[38,49], but also the facilitation of experience-dependent plasticity, such as has been demonstrated for e.g., social/mental training[85].

## Methods

### Study sample

This study included 2245 adolescents and young adults aged 14 to 26 years (54% females; mean age = 19.06 ± 3.02 y) from the NeuroScience in Psychiatry Network (NSPN)[86]. Participants were recruited in Cambridgeshire and north London in an accelerated longitudinal sampling design which balanced sex, ethnicity, and participant numbers in five age strata (14–15, 16–17, 18–19, 20–21, 22–25). All 2245 individuals were included in behavioral analyses (see Supplementary Fig. S10 for details on included sub-samples). Participants' sex was determined based on self-report. This study was conducted in accordance with U.K. National Health Service research governance standards and participants provided informed written consent during NSPN data acquisition, for which ethical approval was granted by the Cambridge East Research Ethics Committee under REC 12/EE/0250. Participants received monetary compensation for their participation.

Our neuroimaging analyses of group-level developmental principles were based on a subsample ($n = 199$; 416 sessions) of adolescents who were invited to undergo longitudinal functional and structural neuroimaging assessments at baseline and a 1 year follow up, with a subsample of 26 subjects invited for an intermediate six months scan. Neuroimaging analyses studying intra- and inter-individual differences with respect to adaptivity were restricted to participants who had at least two structural (MT) and functional scans after quality control, and completed all questionnaires included in this study at two or more timepoints ($n = 141$; 346 sessions; age stratification at baseline: $n = 34/34/22/37/14$; 50.3% female; inter-scan interval = 1.26 ± 0.33 y; Supplementary Fig. S10).

### Generation of resilient psychosocial functioning ($Res_{PSF}$) scores

$Res_{PSF}$ scores were generated in a three-step process (Fig. 1A): (1) Computation of a general distress score ($n = 1533$), (2) prediction of psychosocial distress from environmental adversity measures ($n = 712$), and (3) extractions of residuals from the model for participants with repeated MRI and behavioral data ($n = 141$). To avoid leakage, steps 1 and 2 were based on independent subsamples (Fig. 1A and Supplementary Fig. S10).

Psychosocial distress scores were based on self-report questionnaires spanning mental health domains for which emotional and behavioral symptoms tend to emerge during adolescence and are associated with commonly diagnosed mental disorders. Following previous work on latent mental health dimensions in NSPN[39], the following mental health domains and questionnaires were included: Depression (33-item Moods and Feelings Questionnaire; MFQ[87]), generalized anxiety (including measures of social concerns, worry, physiological change; 28-item Revised Children's Manifest Anxiety Scale; RCMAS[88]), antisocial behaviors (11-item Antisocial Behavior Questionnaire; ABQ), obsessive compulsive behavior (11-item Revised Leyton Obsessional Inventory; r-LOI[89]), self-esteem (10-item Rosenberg Self-Esteem Questionnaire; RSE[90]), psychotic-like experiences (Schizotypal Personality Questionnaire; SPQ[91]), and mental well-being (14-item Warwick-Edinburgh Mental Well-Being Scale; WEMWBS[92]). Please see Supplementary Information for details on the questionnaires. We applied a factor analysis (Matlab 2022b) aiming to derive one latent factor in 1533 individuals. This latent factor correlated highly ($r = 0.99$) with the general distress score derived from previously reported Bi-factor models that additionally include five sub-factors[39]. In a separate subsample ($n = 712$), we then applied the derived item loadings to each individual's respective item scores. The sum of item scores multiplied by item loadings defined subject-level distress scores.

Conceptualizing relatively more resilient or susceptible outcomes as lower or higher than expected distress, respectively, given the

adversity faced, we then predicted distress scores from available adversity measures. These included: The Life Events Questionnaire (LEQ[93]), Child Trauma Questionnaire (CTQ[94]), Alabama Parenting Questionnaire (APQ[95]), Measure of Parenting Style (MOPS[96]), and socioeconomic status (as approximated by zip codes/IMD). See Supplementary Methods for details on included questionnaires. For the prediction, we used a random forest regression in a supervised machine learning approach implemented in sci-kit learn (v1.2.1, https://scikit-learn.org, in Python v3.10.9). We applied a nested cross-validation in which we left all sessions of one subject out in the outer scheme, i.e., 712 outer folds, and split the remaining data into five even groups for training, i.e., five inner folds, in each iteration. Performance was estimated based on mean absolute errors and parameter optimization was performed for the number of estimators (50, 100, 150, 200, 250, 300) and tree depth (5 to 15). We included a StandardScaler (z-scoring) to preprocess features within the cross-validation scheme.

### Neuroimaging data acquisition
Magnetic Transfer (MT) data was acquired as a neuroimaging proxy of myelin content using a multi-parametric mapping (MPM) sequence[23] on three identical 3 T Siemens MRI Scanners (Magnetom TIM Trio) in Cambridge (2) and London (1). A standard 32-channel radio-frequency (RF) receive head coil and RF body coil for transmission were used. Anatomical and functional data were acquired on the same day. Neuroimaging data acquisition and processing has also been described previously[45,97].

### Myelin-sensitive MRI
MPM comprised three multi-echo 3D FLASH scans: predominant T1-weighting (repetition time (TR) = 18.7 ms, flip angle = 20°), and predominant proton density (PD) and MT-weighting (TR = 23.7 ms; flip angle = 6°). To achieve MT-weighting, an off-resonance Gaussian-shaped RF pulse (duration = 4 ms, nominal flip angle = 220°, frequency offset from water resonance = 2 kHz) was applied prior to the excitation. For MT weighted acquisition, several gradient echoes were recorded with alternate readout polarity at six equidistant echo durations (TE) between 2.2 and 14.7 ms. The longitudinal relaxation rate and MT signal are separated by the MT saturation parameter, creating a semi-quantitative measurement that is resistant to field inhomogeneities and relaxation times[23,98]. Further acquisition parameters: 1 mm isotropic resolution, 176 sagittal partitions, field of view (FOV) = 256 × 240 mm, matrix = 256 × 240 × 176, parallel imaging using GRAPPA factor two in phase-encoding (PE) direction (AP), 6/8 partial Fourier in partition direction, non-selective RF excitation, readout bandwidth BW = 425 Hz/pixel, RF spoiling phase increment = 50°. The acquisition time was approx. 25 min, during which participants wore ear protection and were instructed not to move and rest. MPM further comprises a set of other contrasts, such as R2* sensitive to iron content, yielding complementary insights into different aspects of tissue micro-architecture in vivo[49,99–101]. Here, we focused on MT, which is considered a particularly strong in vivo marker of myelin with a high spatial correspondence with myelin basic protein and other myelin-related molecules in the brain, as has been verified by several histological validation studies[24–27]. MT has further been demonstrated to show high reliability[102] suitable for the study of individual differences and brain-behavior associations[5,16,100].

### Resting-state functional MRI
Resting-state functional MRI (fMRI) data were acquired using a multi-echo echo-planar imaging sequence (TR = 2.42 s; GRAPPA with acceleration factor = 2; flip angle = 90°; matrix size = 64 × 64 × 34; FOV = 240 × 240 mm; in plane resolution = 3.75 × 3.75 mm; slice thickness = 3.75 mm with 10% gap, sequential slice acquisition, 34 oblique slices; bandwidth, 2368 Hz/pixel; TE = 13, 30.55, and 48.1 ms).

### Neuroimaging data preprocessing
**Microstructure.** T1w and MT images were visually inspected for motion artifacts (such as ringing, ghosting, smearing or blurring) by experts and scans were strictly excluded if motion artifacts were detected. Surface reconstruction was performed on T1w data using the Freesurfer _recon-all_ command (v.5.3.0[103]). Briefly, the pipeline performs non-uniformity correction, projection to Talairach space, intensity normalization, skull stripping, automatic tissue segmentation, and construction of the gray/white interface and the pial surface. Surface reconstructions/segmentations were edited by adding control points in FreeSurfer, re-processed, and then underwent quality control again. If further motion artifacts were detected in this process, the relevant scans were excluded. MT images were co-registered with reconstructed surfaces and 12 equivolumetric cortical surfaces were generated within the cortex (i.e., between the pial and white surface[104]). The equivolumetric model takes cortical folding into account by manipulating the Euclidean distance (ρ) between intra-cortical surfaces, thereby preserving the fractional volume between pairs of surfaces (1):

$$\rho = \frac{1}{A(\text{out}) - A(\text{in})} \times \left( A(\text{in}) + \sqrt{\alpha A^2(\text{out}) + (1 - \alpha)A^2(\text{in})} \right) \quad (1)$$

$\alpha$ = a fraction of the total volume of the segment accounted for by the surface; $A(\text{out})$ and $A(\text{in})$ = the surface areas of outer and inner cortical surfaces, respectively.

The outer two surfaces were excluded to avoid potential partial volume effects (PVE) and MT intensities were extracted from 10 cortical depths at each vertex. In addition, depth-specific PVEs caused by cerebrospinal fluid (CSF) were corrected for using a mixed tissue class model[105]. To this end, a linear model was fitted to each node at all 10 depths (2):

$$MT_{(n,s)} \sim b_0 + b_1 CSF_{(n,s)} \quad (2)$$

where $n$ = node, $s$ = surface. Derived CSF-corrected MT values reflect the sum of residuals (3):

$$MT_{c(n,s)} = T1_{(n,s)} - (b_0 + b_1 CSF_{(n,s)}) \quad (3)$$

and original group averaged MT.

Last, vertices were averaged within 360 bilateral cortical parcels using the Human Connectome Project (HCP) parcellation atlas that was mapped from standard fsaverage space to each participant's native space using surface-based registration[104,106].

### Resting-state functional MRI
Multi-echo independent component analysis (ME-ICA[107,108]) was applied to the fMRI data to isolate and remove variance caused by sources that do not scale linearly with the TR within the time series and are therefore assumed not to represent the blood oxygenation level dependent (BOLD) contrast. Variance in cerebrospinal fluid was estimated based on ventricular time series and regressed from parenchymal time series via Analysis of Functional NeuroImages (AFNI[109]). Functional data were co-registered to R1 images, which were derived from the same MPM sequence as MT data, ensuring spatial alignment between functional and MT data. Volumes obtained within the 15-s steady-state equilibration were excluded. Anatomical-functional co-registration and motion correction parameters were computed using the middle TE data, and the base EPI image was the first volume following equilibration. Matrices for de-obliquing and six-parameter rigid body motion correction were computed. Using the LPC cost function with the EPI base image as the LPC weight mask, a 12-parameter affine anatomical-functional co-registration was computed. Matrices for de-obliquing, motion correction, and anatomical-functional co-registration were concatenated into a single alignment matrix using the AFNI

tool align_epi_anat.py. The dataset of each TE was then slice-time corrected and spatially aligned through repeated application of the alignment matrix. Data was parcellated into the same 360 bilateral HCP cortical regions applied to structural data within which regional time series were averaged across voxels of each respective parcel. Moreover, a band-pass-filter (range: 0.025 to 0.111 Hz[110]) was applied to the regional time series using discrete wavelet transform. Following quality control, regional time series were z-scored and 30 regions, mainly in paralimbic areas, were excluded due to low z-scores ($Z < 1.96$) in at least one participant. A functional connectivity (FC) matrix was generated for each subject by computing Pearson's correlation coefficients between all 330 remaining parcels, yielding a $330 \times 330$ matrix. Correlation coefficients were then z-transformed by Fisher's transformation[111]). Hence, FC units represent standard deviations of the normal distribution. Lastly, to avoid any residual effects of motion on FC, each edge/Z-score was regressed on each participant's mean frame-wise displacement. All further analyses were based on derived motion-corrected Z-scores (i.e., the residuals of this regression).

In total, 36 scans were excluded due to high in-scanner motion [mean framewise displacement (FD) > 0.3 mm or maximum FD > 1.3 mm], poor surface reconstructions, co-registration errors, and/or extensive fMRI dropout.

### Intra-individual change
Intra-individual change in mean and layer-wise MT, as well as resting-state functional connectivity (FC), was assessed by calculating the Δ between first and last MRI sessions: $MT_{T2}-MT_{T1}$ or $FC_{T2}-FC_{T1}$ respectively, for each parcel. Δs were winsorized to +/− 3 SD to account for outliers.

### Association between myeloarchitectural and intrinsic functional maturation and change in resilient psychosocial functioning
The association between change in $\Delta RES_{PSF}$ and ΔMT was assessed by applying a general linear model to each parcel (4):

$$\Delta MT(parcel) \sim 1 + \beta_{\Delta Res_{PSF}} * \Delta Res_{PSF} + \beta_{meanRes_{PSF}} * mean\,Res_{PSF} \\ + \beta_{age} * age + \beta sex * sex + \beta_{site} * site + \epsilon \qquad (4)$$

Models were fitted in SurfStat[112] (Matlab 2022b) adjusting for mean age (i.e., $(Age_{baseline} + Age_{follow\,up})/2$) and mean $Res_{PSF}$ (i.e., $(RES_{PSF\,at\,baseline} + RES_{PSF\,at\,follow\,up})/2$) across sessions, sex, and site. Before fitting the model, we adjusted $\Delta Res_{PSF}$ and ΔMT for inter-session-intervals, which varied between participants, by fitting linear regressions of Δage. This was done separately for imaging and behavioral data to adjust for the fact that behavioral and imaging data were mostly not collected on the same day. All statistical tests reported throughout this study comprised two-sided testing. Significance was assessed by non-linear permutation testing (10.000 permutations of Δ and mean $Res_{PSF}$ and FDR correction of derived p-values at α < 0.05). We then stratified the unthresholded t-map according to a cytoarchitectonic map defining six dominant cortical types[28], to reveal potential systematic links between cortical architecture and effects associated with changes in resilient psychosocial functioning.

We ran two post-hoc analyses based on the region of interest (ROI) defined by parcels that show a significant $\Delta Res_{PSF} *$ΔMT association. First, we tested whether significant associations between $\Delta Res_{PSF}$ and ΔMT were layer-specific by fitting the same linear model to MT values at each of the 10 surfaces, rather than the mean across surfaces. This was done to reveal a potential specificity of effects based on cortical depth, not to statistically confirm the observed association again. Next, we addressed the question of whether regions showing differences in ΔMT as a function of $\Delta Res_{PSF}$ also exhibit different functional connectivity. To this end, we computed global FC of the ROI as degree centrality (i.e., the sum of all connections) as well as a seed-

based FC analysis defining the ROI as a seed. As described for the ΔMT analysis, we regressed out the effects of inter-session intervals, fitted the same linear model as described above, and assessed significance of the seed-based analysis by non-linear permutation testing (10,000 permutations of Δ and mean $Res_{PSF}$ + FDR correction of derived p-values at α < 0.05). Last, we stratified the resulting t-map according to the Yeo 7 Network Atlas[47] to reveal systematic effects within specific intrinsic functional networks.

### System-level maturation
We studied system-level maturation based on both myeloarchitectural and functional data. In order to assess the cortical topology of MT maturation, we computed a microstructural profile covariance (MPC) matrix and studied its change with age. MPC is based on myeloarchitectural profiles across cortical depths and is generated via partial correlations between nodal MT profiles of two given regions, corrected for the mean MT intensity across intra-cortical surfaces (Fig. 3A). An underlying assumption of this approach is that inter-regional similarity predicts axonal cortico-cortical connectivity[29,33,113]. MPC has previously been shown to align well with post-mortem assessments of inter-regional microstructural similarity[30], and depth-dependent shifts in cytoarchitectonic features such as cell densities or myelin characteristics have been linked to architectural[114] complexity and cortical hierarchy[115].

First, we computed a microstructural and functional Maturational Index (MI), which has been shown to be a robust marker for adolescent modes of reorganization[45] sensitive to individual differences[97] in functional connectivity. To compute the MI, LMEs were fitted to each edge of the MPC and FC matrices, assessing effects of age and including sex, site, and repeated measures of the same individual in the model (5):

$$MRI(k,j) \sim 1 + \beta_{age} * age + \beta_{sex} * sex + \beta_{site} * site + \gamma_{subject} * (1|subject) + \epsilon \qquad (5)$$

Where k and j are two nodes of the matrix. The MI captures the signed Spearman's correlation between predicted baseline patterns, at age 14, and rate of change, age 14–26, of all edges that connect a given node to all other nodes. Thus, it reflects a reorganization of network embedding.

Baseline values at age 14 for each group ($+\Delta Res_{PSF}$ /$-\Delta Res_{PSF}$), $MRI_{14\,group}$, were extracted for MPC and FC matrices as follows (6):

$$MRI_{14} = 1 + \beta_{age} * 14 + \beta_{age} * 14 + \beta_{sex} * (1/2) + \beta_{site1} * (1/3) + \beta_{site2} * (1/3) \qquad (6)$$

Whereas the rate of change $MPC_{14-26}$ or $FC_{14-26}$ simply reflects the β-coefficient of age (7):

$$MRI_{14-26}(k,j) = \beta_{age} \qquad (7)$$

At each node, we then computed the row-wise Spearman's $\rho$ between ranked extracted parameters reflecting baseline and change parameters of all edges (i.e., 360 for MPC, 330 for FC) of a specific node. A positive correlation indicates that a given region's edges that were already similar in either their myeloarchitectural or functional profile became more similar with development, this is termed 'conservative' development. Conversely, a negative correlation reflects reorganization, edges that were similar at baseline differentiate, or edges that were dissimilar at baseline integrate, which is termed 'disruptive' development. We computed normative MIs for MPC and FC across all participants ($n = 199$) using all existing sessions (416 sessions). To probe convergence and divergence of structural compared to functional maturational modes, we further tested overlaps between regional MIs (individually thresholded at pFDR <0.05).

Next, we contextualized the $MI_{MPC}$ with a previously established measure of global organization of microstructural maturation: The MPC principal axis of age effects[16]. To this end, we applied the same LME as defined in Eq. 5 to each edge of the MPC matrix, assessing the main effect of age on inter-regional microstructural similarities. Diffusion map embedding was then applied to the matrix of $t$-values (thresholded at 90%), revealing a cortex-wide organizational axis of synchronized age effects. Regions with a similar loading on this axis are similarly embedded in a network of inter-regional synchronization of age effects, whereas regions at the apices of the axis show maximally different change patterns.

### Group-level differences in system-level maturation

Last, we aimed to study differences in system-level maturation associated with intra-individual changes in adaptivity. Because the MI is computed from parameters extracted from group-level general linear models, it was required to split the sample into two groups. Thus, the sample was divided into adolescents who showed increasingly resilient outcomes (+ΔRes_PSF; $n = 81$, 193 sessions; 48% female, 18.93 ± 2.81 y at baseline) and adolescents who became more susceptible with age (-ΔRes_PSF $n = 60$; 153 sessions; 53% female, 18.84 ± 2.87 y at baseline). MIs were computed separately for each group, and the resulting maps were subtracted from each other (+ΔRes_PSF - (-ΔRes_PSF)). Significance was tested using two approaches that were combined for thresholding: (1) We first applied Z-tests testing for significant differences between the correlation coefficients, i.e., the difference between group MIs divided by the SE of the difference in MIs, as has been done previously[97] (8).

$$z = \frac{MI_{+Res\_PSF} - MI_{-\Delta Res\_PSF}}{SE_{MI_{+\Delta Res\_PSF} - MI_{-\Delta Res\_PSF}}} = \frac{MI_{+\Delta Res\_PSF} - MI_{-\Delta Res\_PSF}}{\sqrt{SE_{MI_{+\Delta Res\_PSF}}^2 + SE_{MI_{-\Delta Res\_PSF}}^2}} \quad (8)$$

Derived p-values were FDR-corrected at pFDR <0.05.

(2) Next, to control for the effects of sampling bias and potential effects of differences in group size or demographics, we performed non-parametric permutation testing, by shuffling group allocation 10,000 times while considering age and sex distributions as well as group size differences (see Supplementary Fig. S11). Finally, we depicted group differences as significant only if they were significant in both FDR-corrected $p$-values derived from $Z$-tests ($p < 0.05$), and non-parametric permutation testing ($p < 0.05$).

### Reporting summary

Further information on research design is available in the Nature Portfolio Reporting Summary linked to this article.

## Data availability

The behavioral resilience scores and microstructural profiles generated in this study have been deposited on Github and Zenodo under https://github.com/CNG-LAB/cngopen/tree/main/adolescent_resilience/ScrFun and https://zenodo.org/records/11486553. The item-level questionnaire data as well as unprocessed imaging data can be obtained from https://portal.ide-cam.org.uk/overview/6/managed or https://www.repository.cam.ac.uk/handle/1810/264350. The processed functional connectivity data are available at https://zenodo.org/records/6390852. The depicted data generated in this study are provided in the Supplementary Information/Source Data file. Source data are provided with this paper.

## Code availability

Custom code generated for this project was made publicly available under https://github.com/CNG-LAB/cngopen/tree/main/adolescent_resilience/ScrFun and https://zenodo.org/records/11486553. Our analysis code makes use of open software: Gradient mapping analyses were carried out using BrainSpace (v. 0.1.2; https://brainspace.readthedocs.io/en/latest/) and surface visualizations were based on code from the

ENIGMA Toolbox (v.1.1.3; https://enigma-toolbox.readthedocs.io/en/latest/[116]) in combination with ColorBrewer (v. 1.0.0; https://github.com/scottclowe/cbrewer2), and the Violin Plot Toolbox (Holger Hoffmann (2024); https://www.mathworks.com/matlabcentral/fileexchange/45134-violin-plot). Statistical analyses were carried out using SurfStat (https://www.math.mcgill.ca/keith/surfstat/). Equivolumetric surfaces were computed using code from: https://github.com/MICA-MNI/micaopen/tree/master/a_moment_of_change[16]. Z-tests were performed using the compare correlation coefficients function (Sisi Ma (2024). compare_correlation_coefficients (https://www.mathworks.com/matlabcentral/fileexchange/44658-compare_correlation_coefficients). *Python*: We made use of the following packages: scipy 1.10.1, sklearn 0.0.post1, matplotlib 3.7.1, numpy 1.24.2, pandas 1.5.3, seaborn 0.11.0.

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

## Acknowledgements

Many scientists contributed to the NSPN consortium but did not take active part in the writing of this report. A full list of contributors to NSPN is available in the Supplementary Material. The authors would like to express their gratitude to the open science initiatives that made this work possible. MDH was funded by the German Federal Ministry of Education and Research (BMBF) and the Max Planck Society. L.D. was supported by a Gates Cambridge Scholarship. S.L.V. was supported by the Max Planck Society through the Otto Hahn Award and the Helmholtz International BigBrain Analytics and Learning Laboratory (Hiball). Data were curated and analysed using a computational facility funded by an MRC research infrastructure award (MR/M009041/1) to the School of Clinical Medicine, University of Cambridge and supported by the mental health theme of the NIHR Cambridge Biomedical Research Center. The views expressed are those of the authors and not necessarily those of the NIH, NHS, the NIHR or the Department of Health and Social Care.

## Author contributions

The authors confirm contribution to the paper as follows: Study conception and design: M.D.H. and S.L.V. Data collection: E.T.B. and NSPN. Analysis and interpretation of results: M.D.H., S.L.V., L.M.J., and L.D. Draft paper preparation: M.D.H. and S.L.V. Draft paper revision: M.D.H., S.L.V., S.B.E., L.D., L.P., L.M.J., C.P., R.A.B., and E.T.B. All authors reviewed the results and approved the final version of the paper.

## Funding

## Competing interests

E.T.B. works in an advisory role for Sosei Heptares, Boehringer Ingelheim, GlaxoSmithKline, and Monument Therapeutics. E.T.B. is a stockholder and director of Centile Bioscience Ltd. RAB is a director of and hold equity in Centile Bioscience Ltd. The remaining authors declare no competing interests.

## Additional information

## NSPN Consortium

**Edward T. Bullmore**[5]

A full list of members and their affiliations appears in the Supplementary Information.

