## [Peer Review File · Nature Communications]

Longitudinal variation in resilient psychosocial functioning is associated with ongoing cortical myelination and functional reorganization during adolescenceREVIEWER COMMENTS

Reviewer #1 (Remarks to the Author):

Thank you for the opportunity to review this manuscript. This work examines intra-individual adolescent psychosocial functioning relative to adversity exposure, finding associations with myeloarchitectural and functional features assessed at two time points, baseline and follow up. The study questions are novel and contribute to the literature in informing thinking of functional and white matter structural development in the context of resilience in adolescence and early adulthood. I have some comments:

The study lacks discussion of effect sizes and magnitude of effects throughout. Consistent reporting of effect sizes in addition to interpreting results in light of effect sizes is warranted.

Two timepoints are not enough to reliably detect change (e.g., <https://www.google.com/url?sa=t&rct=j&q=&esrc=s&source=web&cd=&ved=2ahUKewis3e6NjPWEAxXJIYkEHWzXBesQFnoECA8QAw&url=https%3A%2F%2Fosf.io%2F96ph3%2Fdownload%23%3A%2F%3Atext=Even%2520under%2520ideal%2520simulated%2520conditions%2Ctime%2520points%2520across%25201000%2520samples.&usg=AOvVaw1I95hyh10JN3ZNP6aCAKRv&opi=89978449>). I would hesitate to say these reflect “maturational trajectories” given only two observations. More discussion and acknowledgement is warranted.

I would appreciate more discussion on to what extent the residuals reflect degrees of resilience, and to what extent they could reflect noise or measurement error.

At some points it is not fully clear what the sample size for each subset of analyses was: for some it seems it was as low as 200 or less. Could the authors comment on possible challenges related to replicability in neuroimaging work with such sample sizes (e.g., <https://doi.org/10.1038/s41586-022-04492-9>)? Some careful discussion is warranted.

SES is one of the three measures incorporated into adversity exposure. Could there be concerns with restriction of range, given that higher SES is disproportionately represented in individuals who participate in neuroimaging research? Relatedly, a bit more discussion on representativeness of this sample relative to the population it intends to represent would be helpful.

The manuscript would benefit from some editing for clarity. For example, the abstract does not note across what timeframe subjects were observed. Terms like “mesoscale maturation” are used in the introduction without definition, which are not likely to be readily recognizable to broader audiences without an intensive neuroimaging focus.

Reviewer #2 (Remarks to the Author):

In the manuscript “Longitudinal trajectories of resilient psychosocial functioning link to ongoing cortical myelination and functional reorganization during adolescence” Hewett et al examined data from two time points in individuals 14-26 years old (N=141). Specifically, they looked at 1) how changes in stressor resilience scores corresponded to changes in magnetization transfer (MT), 2) the extent to which this effect was homogenous across differing cortical depths, and whether these changes are linked to certain cytoarchitecture cortical types, 3) how prefrontal functional connectivity (seed region from aim 1) change was associated with social resilience score change, and 4) how multimodal systems level maturation was related to one another and 5) how this systems level maturation was related to change in stressor resilience scores. Results were: 1) There was widespread increased myelination over time in the sample and increased myelination in the anterolateral prefrontal cortex was associated with more resilient psychosocial functioning, and 2) the effect was homogenous across differing cortical depths and localized to Eulaminate Cortex II and III, 3) less change/more stabilization in prefrontal connectivity was associated with more resilient functioning, 4) systems level analyses found more “conservative” development in ventral

temporal and dorsal regions (i.e., positive correlation between baseline and change pattern, i.e., integration of regions w/ higher myeloarchitectural similarity at baseline and/or a differentiation of regions that were already dissimilar at baseline) and “disrupted” re-organization in heteromodal fronto-parietal cortex (i.e. regions that were more dissimilar at baseline became more integrated with each other and/or regions with higher myeloarchitectural similarity became more differentiated), and 5) more resilient individuals had less conservative development paired with more “disrupted” development at the systems level. This is an important manuscript that was conducted in a scientifically rigorous method. The results fill an important gap in the literature – specifically examining neurobiological substrates of people who are exposed to multiple adversities and still exhibit adaptive development. The authors truly provide “nuanced and multimodal evidence that maturational trajectories of late-maturing association cortices are associated with changing abilities to adapt to psychosocial stressors during adolescence.” Analyses are sophisticated and creative. Figures are beautiful and help to clearly tell the story of the manuscript. I loved how clearly the Discussion was written and enjoyed reading the thoughtful insights that the authors provided. I have some suggestions below that I feel will improve the quality of the manuscript and I hope that the authors find the feedback helpful. I must admit that I greatly enjoyed reading this manuscript, in fact it may be one of the most well-written + most scientifically informative manuscripts I have read in the last year.

Here is my feedback:

1. For me, it was difficult to understand the exact aims of the study from the abstract and the end of the first paragraph in the Introduction. I was able to figure out the main aims by examining Figure 1 and Figure 2, but it would be helpful if the authors could be a bit more explicit in these two sections.
2. It would also be helpful to have the final sample size (N=141) and number of time points specified in the abstract.
3. It is confusing to have the abbreviation “SRS” used to reflect the stressor resilience score, given that there is a widely used measure of “Social Responsiveness Scale”, a self- and/or parental report that measures autistic traits continuously. I am guessing that the authors chose this abbreviation to reduce their word count. Therefore, I respectfully ask the editor if the authors can increase the word count to account for this increase in words. I found this to be one of the more distracting pieces of the manuscript (constantly re-oriented myself to what SRS stood for in this study).
4. The discussion of how the authors examined how the change in MT – change in social resilience score relationships are linked to “specific cytoarchitectonic cortical types along a histologically defined hierarchy of microstructural complexity” comes out of the blue for me. How did the authors do this? It would be helpful to add a sentence or two here to provide a bit of context. I recognize the methods section is separate, but I was able to understand all of the other methods enough to understand the results, except for this one method.
5. Like there is a section titled “Fundamental patterns of MT maturation in the full imaging sample”, it would be helpful to see a results section on “changes in social resilient social scores” in the imaging sample. A descriptive characterization of this score and how it changes over time would be helpful for a fuller understanding of the student.
6. I became confused when I read this section “....previously established principal axis of age-related change (Figure 3C-E), which captures a differentiation of idiosyncratic sensory vs. paralimbic/temporal maturational patterns”. Could the authors more explicitly describe the exact pattern they are talking about? All I understand from reading this is there is already an age-associated change found, and it differentiates between sensory and temporal regions. However, I don’t have any idea what this pattern looks like and I would like to know that.

Reviewer #2 (Remarks on code availability):

I did not have time to do this. I would need another few days to review the manuscript, and my review was already late.

Reviewer #3 (Remarks to the Author):

This is an elegant and interesting paper using the NSPN consortia data to understand how changes in myelination in adolescence are related to changes in resilient adaptation. The authors report a positive association between left-lateralised anterolateral prefrontal cortex myelination and more resilient functioning. They then investigated changes in functional connectivity using this region as a seed, more segregation in global connectivity patterns was associated with less resilient functioning.

I think the paper is relatively well-written and addresses an issue that is relatively unaddressed in the field. I found the use of a microstructural covariance matrix to understanding similarities in neural profiles interesting and compelling. My chief concerns are that the methods used are complex and valuable, but not enough has been done to show that they are reliable and meaningful (for example, does the maturational index have value beyond this paper?).

I had the following questions for the authors:

1. How do we know the relationship between myelination and psychosocial functioning is specific? Changes in myelination may also be driven by other ongoing changes in adolescence, such as cognitive changes and changes in environments experienced by adolescents, or even third factors such as SES?
2. In the discussion, the implication is that myelination of the prefrontal cortex drives resilient adaptation. However, changes in myelination might reflect the use of these different strategies. Does the longitudinal data help to disentangle this?
3. The number of children with longitudinal MRI data is relatively small (N=141). What size of effects were expected? Was a power analysis conducted prior to data analysis?
4. Why did the authors choose to focus on MTsat from the MPM data, rather than other measures such as R1 or R2*?
5. Did the authors assess the effect of movement on their MPM data in this developmental group? What quality control pipeline was used?
6. How reliable was the prediction of distress scores from adversity measures?
7. How easy was the alignment and registration between the MPM data and the functional data? How was this checked?
8. Although the microstructural covariance matrix argument is compelling, from the paper, it is really hard to understand how this is being calculated, and therefore evaluate further arguments. It would be helpful if this was clearer.

Response to Reviewers (NCOMMS-24-08809)

We would like to thank the Editors and Reviewers for their positive evaluations, constructive comments, and for the opportunity to submit a revised manuscript. We believe that the resulting changes have significantly improved the clarity and quality of our work. We have addressed all questions and suggestions in a point-by-point fashion below and we have edited our manuscript and supplementary material accordingly. We have uploaded the revised files as well as files highlighting changes in yellow.

Reviewer #1

Thank you for the opportunity to review this manuscript. This work examines intra-individual adolescent psychosocial functioning relative to adversity exposure, finding associations with myeloarchitectural and functional features assessed at two time points, baseline and follow up. The study questions are novel and contribute to the literature in informing thinking of functional and white matter structural development in the context of resilience in adolescence and early adulthood. I have some comments:

Thank you for the positive feedback and constructive comments which we have addressed below.

1.1 The study lacks discussion of effect sizes and magnitude of effects throughout. Consistent reporting of effect sizes in addition to interpreting results in light of effect sizes is warranted.

We thank the Reviewer for pointing this out and agree with the comment. We now added standardized betas whenever we reported a t-value in text and acknowledge the size of effects when addressing results.

e.g.

"We observed a positive association between developing towards more resilient functioning (i.e., an intra-individual increase in ResPSF scores) and ΔMT in the predominantly left-lateralized anterolateral prefrontal cortex (PFC; $t_{max}(134) = 4.51$; $\beta_{standardized} = 0.36$ (medium effect); $p_{10,000 \text{ permutations}} \& FDR < 0.05$; Figure 2A).

[...]

Globally, we observed more maintained levels of cortex-wide functional connectivity with increasingly resilient outcomes, whereas increasingly susceptible outcomes were associated with a segregation (small global effect: $t(134) = 2.45$; $\beta_{standardized} = 0.21$; $p = 0.02$).

[...]

Studying the cortex-wide pattern of associations between changes in ResPSF scores and prefrontal ΔFC revealed that effects were concentrated in regions of the default mode, frontoparietal and ventral attention networks (medium regional effect: $t_{max}(134) = 3.78$; $\beta_{standardized} = 0.31$; $p_{10,000 \text{ permutations}} \& FDR < 0.05$; Figure 2C & D)"

For the between-group comparison of maturational indices (correlation coefficients reflecting the correlation between baseline and age-related change patterns), we now show Cohen's q effect sizes directly plotted on the surface maps (please see updated Figure 4A and Supplementary Figure S7).

A | MPC Maturation index in increasingly resilient vs. susceptible psychosocial functioning

Figure 4. More disruptive re-organization of microstructural profile covariance (MPC) networks with increasingly resilient mental health outcomes. A) Top; Cohen's q for group differences in the Maturation index (MI_{MPC}; FDR $p < 0.05$ & 10,000 permutations, $p < 0.05$) indicating a widespread negative shift (i.e., more disruptive or less conservative MPC development) in the group of individuals who developed towards more resilient outcomes ($\uparrow \Delta \text{Res}_{\text{PSF}}$). [...]

1.2 Two timepoints are not enough to reliably detect change (e.g., <https://www.google.com/url?sa=t&rct=j&q=&esrc=s&source=web&cd=&ved=2ahUKEwis3e6NjPW EAxXJIYkEHWzXBesQFnoECA8QAw&url=https%3A%2F%2Fosf.io%2F96ph3%2Fdownload%23%3A~%3Atext=Eve n%2520under%2520ideal%2520simulated%2520conditions%2Ctime%2520points%2520across%252010 00%2520samples.&usg=AOvVaw1I95hyh10JN3ZNP6aCAKRv&opi=89978449>). I would hesitate to say these reflect “maturation trajectories” given only two observations. More discussion and acknowledgement is warranted.

Thank you for raising this concern, we agree. We re-named ‘trajectories’ to ‘longitudinal variation’ or ‘changes’ throughout the manuscript when describing current findings – including the title. We further highlighted this potential limitation, including the reference you shared, in the discussion of study limitations:

“We acknowledge that our analyses of resilient psychosocial functioning are limited to the inclusion of only two timepoints. While this allows us to study longitudinal variation, we point out that more time points per participant would be required to estimate trajectories with higher reliability (Parsons & McCormick, 2024). Future studies utilizing e.g., later release waves of the Adolescent Brain Cognitive Development (ABCD) cohort, may trace longitudinal trajectories over a longer time period and further assess the question whether enhanced myelin maturation is also a predictor for adult mental health (McLaughlin et al., 2020).”

1.3 I would appreciate more discussion on to what extent the residuals reflect degrees of resilience, and to what extent they could reflect noise or measurement error.

Thank you for addressing this point.

We agree that a more thorough discussion of the residual approach is helpful and have now added a new section elaborating on this issue in the Supplementary Information:

“Insights gained and potential sources of noise related to using residuals as resilience scores

Assessing resilient/susceptible outcomes based on the deviation of observed from expected well-being levels given a certain adversity exposure is a well-established approach in the resilience literature (Bowes et al., 2010; Collishaw et al., 2016; Kalisch et al., 2017; Miller-Lewis et al., 2013; Sapouna & Wolke, 2013; Van Harmelen et al., 2017; Veer et al., 2021). It addresses the issue that simple quantification of mental health variables can provide only limited insights into resilience or susceptibility, as they are heavily conflated with individual differences in adversity exposure (Kalisch et al., 2021). Leveraging residuals to quantify better or worse well-being than predicted by adversity exposure thus

provides a corrected well-being score that is adjusted for individual differences in stressor exposure and thus allows comparing resilience scores of individuals with differing exposure levels. That is, ‘residuals’ in the resilience use case are interpreted as ‘residual variance in mental health problems’ that is not explained by the normative response to exposure, therefore indicating individually weaker response (resilience) or greater response (susceptibility). It may thus be compared to the process of correcting a dependent variable for potential confounds such as age or sex.

While the basic assumption of this approach is that residualized mental health outcomes reflect degrees of resilience/susceptibility, there are other potential influences such as 1) measurement error and noise related to the questionnaires used, 2) noise related to the performance of our prediction model, and 3) confounding influences of other environmental influences.

- 1) Self-report / retrospectivity biases may pose one source of measurement error, e.g. for the reporting of childhood maltreatment (Baldwin et al., 2019), despite the relatively short reporting time windows of the current study. It should be noted, however, that such sources of measurement error are not unique to / caused by the residual approach, but rather persist from the original measures of psychosocial well-being. The resilience scores should therefore not contain more measurement error than the original measures of psychosocial well-being. In the current study, our longitudinal approach may further mitigate some aspects of measurement error if it is linked to e.g., self-report bias. That is, an individual systematically self-reporting his/her well-being as a bit better than it is will receive higher resilience scores at all time points. However, as our study investigates intra-individual change, such a general offset in resilience scores would not affect change scores, whereas it would affect cross-sectional analyses more strongly.*
- 2) Another question is how well our prediction model controls for differences in adversity exposure in order to reveal resilient / susceptible responses – and to what extent influence is still uncontrolled for, thus causing noise in in our measure of individual resilience. The amount of variance explained by our model (approx. 21%) is comparable with previous work reporting 24% of variance explained for the prediction of psychosocial functioning from family experiences in longitudinal settings (Van Harmelen et al., 2017); 21% when predicting internalizing symptoms from general life stressors during the covid pandemic (Veer et al., 2021); or 28% when predicting psychosocial functioning from childhood adversity (González-García et al., 2023). We thus conclude that our model controls for exposure to a comparable degree as common resilience models.*
- 3) Capturing meaningful variation in psychological outcomes is complicated by the complex influence of a multitude of interacting factors, which likely contribute to the variance not explained by our model. Such factors may include genetic predispositions, other environmental risk or protective factors not measured here, but likely also sources of noise we cannot quantify. For instance, an individual with a genetic predisposition for mental illness may systematically show lower psychological well-being, which cannot fully be explained by adversity exposure and would thus create a bias in that individuals’ derived resilience scores. Similar to 1), if general offsets in resilience scores exist due to e.g., a genetic predisposition, studying longitudinal change helps us to account for such an offset.*

Overall, our results using resilient psychosocial functioning scores suggest a central role of multi-modal prefrontal maturation and more wide-spread re-organization of association cortices for resilience and susceptibility during adolescence, tested against null models via non-parametric permutation. This observation is well in line with previous reports of structural and functional involvement of these brain regions in stress responses and susceptibility/resilience (Eaton et al., 2022; Larsen et al., 2023; Luciana & Collins, 2022; Paus et al., 2008; Sydnor et al., 2021a). We believe the fact that observed associations of residuals / resilience scores with multi-modal measures of cortical maturation survived non-parametric permutation tests, and were found in regions previously suggested by the literature, argues for a dominance of meaningful variance reflected in used scores.”

Moreover, we link to this supplementary section in a brief discussion of the issue in the main text:

“Resilient outcomes are assumed to rely on a multi-modal and multi-faceted construct, acknowledging the environments we live in, but also other psychological variables beyond clinical symptomatology (such as positive affect, life satisfaction, personality traits). We cannot clearly disentangle the interaction of different intrinsic and extrinsic influences contributing to an individual’s psychological well-being beyond resilience. While we estimated resilience/susceptibility by adjusting psychosocial well-being for adversity exposure - yielding a residualized psychosocial functioning score - we cannot rule out that derived resilience scores also reflect influences of other genetic and environmental factors, and noise or measurement error to a certain degree (see Supplementary Information). For instance, potential self-report / retrospectivity biases inherent to measures of well-being and adversity exposure (Baldwin et al., 2019) would persist in resilience scores, but are not caused by the residual approach providing resilience scores. Our longitudinal approach allowed us to limit confounding effects of individual differences in genetic predispositions or environmental circumstances (such as family composition or neighborhood) that are likely to contribute less variability to within-subject repeated measures than to between-subject cross-sectional data. Overall, our model (explaining 21% of variance) controls for exposure to a comparable degree as common resilience models (explaining 21-28% of variance; see e.g., (González-García et al., 2023; Van Harmelen et al., 2017; Veer et al., 2021). We further aimed to increase robustness and generalizability of the model by implementing a nested cross-validation and a random forest regression robust to non-linear and non-parametric distributions of questionnaire data.”

1.4 At some points it is not fully clear what the sample size for each subset of analyses was: for some it seems it was as low as 200 or less. Could the authors comment on possible challenges related to replicability in neuroimaging work with such sample sizes (e.g., <https://doi.org/10.1038/s41586-022-04492-9>)? Some careful discussion is warranted.

Thanks for this remark! We agree that the final sample size of 141 individuals should further be discussed in the discussion. To address our research question, we required: 1. A dataset covering the age range during which many psychiatric symptoms emerge (adolescence/young adulthood). 2. Repeated imaging using 3. Multi-modal sequences, specifically a high-standard myelin proxy, as well as 4. In-depth phenotypic characterization including both measures of adversity exposure and mental health outcomes. There was only one available dataset which satisfied all these criteria and, therefore, we unfortunately did not have a separate sample available for an out-of-sample replication / validation of results.

Though not replicated in an independent sample, we note that our results align very well with the existing literature, highlighting the central role of the prefrontal cortex in stress adaptivity and vulnerability (Eaton et al., 2022; Luciana & Collins, 2022), as well as the role of association cortex maturation in psychiatric susceptibility (Larsen et al., 2023; Paus et al., 2008; Sydnor et al., 2021a). We further observed convergence across modalities, such that, for instance, effects on prefrontal functional network maturation substantively corroborated the observed differences in rates of prefrontal myelination, making it less likely that this finding is caused by noise.

Nonetheless, in an additional effort to address the robustness and generalizability of our results, we performed sub-sampling with replacement and repeated our analyses 1,000 times based on 80% of the data. The resulting patterns were highly correlated with the results based on the full sample (Pearson’s $r = 0.85-0.93$; Supplementary Figures S3 and S8).

Yet, we agree that the present sample size requires a future replication of our results and demands a degree of caution when interpreting the current findings. We have highlighted this issue in our discussion of study limitations:

“We further acknowledge that this sample is of respectable but not massive size. The current sample size resulted from the inclusion of a sample for which adolescent, longitudinal, and multi-modal imaging including a myelin proxy, as well as in-depth phenotypic characterization, was available – rather than an a-priori power-analysis. Overall, we observed that the reported results were robust to model parameter manipulation and sub-sampling, and well in line with the existing literature, highlighting the

central role of the PFC in stress adaptivity and vulnerability (Eaton et al., 2022; Luciana & Collins, 2022), as well as the role of association cortex maturation in psychiatric susceptibility (Larsen et al., 2023; Paus et al., 2008; Sydnor et al., 2021a). Moreover, individual differences in rates of myelination were observed in regions generally showing highest rates of myelination during adolescence (Figure 1B). This supports a link to the protracted critical period of plasticity in the prefrontal cortex, extending well into early adulthood and associated with increased susceptibility to environmental risk factors (Larsen et al., 2023). We further observed convergence across imaging modalities, for example, effects on prefrontal functional network maturation substantively corroborated the observed differences in rates of prefrontal myelination. However, recent reviews of the replicability of neuroimaging studies linked to insufficient sample sizes (Marek et al., 2022) have warned about inflated effect sizes. Therefore, we stress that current results should be interpreted with caution, pending replication in future studies including independent and larger samples.”

1.5 SES is one of the three measures incorporated into adversity exposure. Could there be concerns with restriction of range, given that higher SES is disproportionately represented in individuals who participate in neuroimaging research? Relatedly, a bit more discussion on representativeness of this sample relative to the population it intends to represent would be helpful.

We thank the Reviewer for raising this concern which has prompted us to assess the general representativeness of the imaging subsample with respect to both levels of risk exposure (including SES) and well-being in a new section of Supplementary Information. While we do indeed see a higher skewness of the SES distribution in the imaging sub-sample (not in other adversity measures), we have taken much care to ameliorate its effects on our outcome measures, carefully address this in a new supplementary figure, and have now expanded on this point in the Discussion section. We would make three main points in response to this comment: 1) the recruitment strategy, 2) trends observed in the final sample size, and 3) the role this may play in the generation of the behavioral outcome measures.

- 1) *Recruitment:* The NSPN sample was recruited from schools, colleges, NHS primary care services and direct advertisements in north London and Cambridgeshire according to an age-stratified and sex-balanced design. The recruitment of 2245 individuals from different educational and health care sites aimed to engage a wide range of individuals. If we assume that the over-sampling of individuals with higher SES (and potentially lower stressor exposure in general) is concentrated on neuroimaging data, we can assess whether there are systematic differences in the imaging sub-sample compared to individuals who participated only in the behavioral part of the NSPN study, which we did below.
- 2) *Data:* To transparently address and report potential offsets in stressor exposure in the imaging group, we visualized distributions of mental well-being related outcome measures (distress scores and resilient psychosocial functioning scores) and measures of stressor exposure (APQ, MOPS, LEQ, CTQ and SES) in the final imaging sample (n=141) and all other individuals not included in the final analysis (the ‘Non-imaging sample’). We observed strong overlaps between samples for all outcome measures, indicating no systematic offset in measured well-being. We further observed strong sample overlaps in risk assessments related to parenting styles (MOPS and APQ) and life events (LEQ and CTQ). For SES, we observed a similar distribution /range in both imaging and non-imaging sub-samples but a higher skewness indicating relatively more individuals with a higher SES in the imaging sub-sample compared to the non-imaging sub-sample. We have now included a new section of Supplementary Information reporting these observations (see section ‘*Representativeness of the MRI subsample*’) as well as a new Supplementary Figure to visualize potential offsets in stressor exposures and outcome measures (see Supplementary Figure S2, pasted below).
- 3) *Analysis:* We believe the stronger skewness towards higher SES in the imaging sample had little effect on potential differences in well-being related outcome measures. Resilient psychosocial functioning scores, the main behavioral variable used in our principal analyses, were computed for all individuals for whom risk assessments were available (n=712, including both ‘imaging’ and ‘non-imaging’ subsamples). Therefore, the full distribution of SES was available for this analysis. Moreover, SES had the lowest feature importance (0.1) in the prediction of distress scores from which resilient psychosocial functioning scores were generated, such that its influence on the final behavioral scores was relatively limited compared to that of other adversity measures. However, it may still be the case that the relatively higher frequency of individuals with higher SES

in the imaging sub-sample has an effect at the brain level (Buckley et al., 2019), and may therefore have influenced observed brain-behaviour associations. We have now acknowledged this limitation and included this concern in the Discussion section on study limitations.

Regarding representativeness beyond SES and other measured stressor exposures, the NSPN sample is a locally collected sample from London and Cambridgeshire. 75% of the full sample and 84% of the imaging sub-sample were Caucasian. Moreover, the sample included mostly healthy individuals, thereby excluding individuals with e.g., neurodevelopmental disorders that are part of the general population. We acknowledge that current results would benefit from being replicated in a more inclusive sample that is more representative of the overall population.

We added the following section and Figure to the Supplementary Materials:

“Representativeness of the MRI subsample

To address the concern that developmental neuroimaging studies may not represent the general developmental population (Garcini et al., 2022), we descriptively compared the distributions of levels of environmental risk exposures as well as behavioral outcome measures in $n=144$ individuals included in neuroimaging analyses (the imaging sub-sample) and the larger sample of individuals included in behavioral analyses only (the non-imaging sub-sample; Supplementary Figure S2). We generally observed strongly overlapping distributions for distress and resilient psychosocial functioning scores, as well as for APQ, MOPS, LEQ and CTQ questionnaires, between the two sub-samples. For SES, regarded here as an index of mean deprivation, we observed a comparable range but a relatively higher proportion of higher SES in the imaging sub-sample (i.e., a more left-skewed distribution). This points towards a potential over-sampling of individuals from higher socioeconomic backgrounds for the neuroimaging analysis. At the same time, it should be noted that the behavioral outcome measure (resilient psychosocial functioning, Res_{PSF}) was computed in the full sample, including more individuals with lower SES, and also that SES was weighted with lowest feature importance by the model used to predict Res_{PSF} (Figure 1A). We also note that the NSPN sample is a locally collected sample from London and Cambridgeshire in the UK. 75% of the full sample and 84% of the imaging sub-sample were Caucasian. Moreover, the sample includes mostly healthy individuals, thereby excluding individuals with e.g., neurodevelopmental disorders that are part of the general population. It will be important to test the replicability of the current results in more socio-economically and ethnically representative samples of the general population. The effectiveness of potential resilience factors implied by the current study in a largely Caucasian, relatively affluent and healthy sample requires further validation in ethnically, socio-economically and clinically defined groups that are under-represented in this sample.

A | Distress and Stressor resilience scores in MRI and behavioural subsamples

B | Stressor exposure in MRI and behavioural subsamples

Supplementary Figure S2. Representativeness of the MRI subsample. The MRI subsample comprises the 141 individuals included in the main analyses linking changes in resilient psychosocial functioning (Res_{PSF}) to myeloarchitectonic and functional maturation. The Non-MRI sample comprises all other individuals that were included in behavioral analyses only and for whom respective behavioral data were available: In A), the non-MRI sub-sample includes $n=314$ individuals with ΔRes_{PSF} scores and $n=885$ individuals with distress scores. In B), the non-MRI sub-sample comprises $n=1457$ individuals with all risk exposure assessments completed. Distributions show density plots. APQ = Alabama Parenting Questionnaire; MOPS = Measure of Parenting Style; LEQ = Life Events Questionnaire; IMD = Index of Mean Deprivation.

And added a link to this new supplementary material in the main manuscript:

“Next, we observed a more strongly skewed distribution of SES in the imaging sub-sample (Supplementary Figure S2), indicating a potential over-sampling of individuals from a comparatively more affluent background. Brain-behaviour associations described here may thus be limited by the reduced variance in SES. This common issue in developmental neuroimaging research (Garcini et al., 2022) demands the study of resilience factors identified by the current study in specific sub-groups, such as cohorts facing specific economic difficulties, that were under-represented in the current sample. However, we also note that distributions in well-being, as well as other risk exposure assessments, did not differ between imaging and non-imaging sub-samples, implying that the over-representation of higher SES participants in the imaging sub-sample was not associated with a commensurate shift in the distributions of risk exposure measures that were weighted more strongly in the prediction of Res_{PSF} .”

[...]

“Therefore, we stress that current results should be interpreted with caution, pending replication in future studies including independent and larger samples, more representative of diverse cultural backgrounds and including individuals typically excluded from healthy samples, such as individuals with neurodevelopmental disorders, to assess the broader generalizability of neurobiological resilience factors identified here.”

1.6 The manuscript would benefit from some editing for clarity. For example, the abstract does not note across what timeframe subjects were observed. Terms like “mesoscale maturation” are used in the introduction without definition, which are not likely to be readily recognizable to broader audiences without an intensive neuroimaging focus.

Thank you for this comment. We removed the term ‘mesoscale’ from the introduction and added the timeframe to the abstract. We further edited and proofread the manuscript, especially the introduction and results section, for clarity and hope it is easier to follow now.

Reviewer #2

In the manuscript “Longitudinal trajectories of resilient psychosocial functioning link to ongoing cortical myelination and functional reorganization during adolescence” Hewett et al examined data from two time points in individuals 14-26 years old (N=141). Specifically, they looked at 1) how changes in stressor resilience scores corresponded to changes in magnetization transfer (MT), 2) the extent to which this effect was homogenous across differing cortical depths, and whether these changes are linked to certain cytoarchitecture cortical types, 3) how prefrontal functional connectivity (seed region from aim 1) change was associated with social resilience score change, and 4) how multimodal systems level maturation was related to one another and 5) how this systems level maturation was related to change in stressor resilience scores. Results were: 1) There was widespread increased myelination over time in the sample and increased myelination in the anterolateral prefrontal cortex was associated with more resilient psychosocial functioning, and 2) the effect was homogenous across differing cortical depths and localized to Eulaminate Cortex II and III, 3) less change/more stabilization in prefrontal connectivity was associated with more resilient functioning, 4) systems level analyses found more “conservative” development in ventral temporal and dorsal regions (i.e., positive correlation between baseline and change pattern, i.e., integration of regions w/ higher myeloarchitectural similarity at baseline and/or a differentiation of regions that were already dissimilar at baseline) and “disrupted” re-organization in heteromodal fronto-parietal cortex (i.e. regions that were more dissimilar at baseline became more integrated with each other and/or regions with higher myeloarchitectural similarity became more differentiated), and 5) more resilient individuals had less conservative development paired with more “disrupted” development at the systems level. This is an important manuscript that was conducted in a scientifically rigorous method. The results fill an important gap in the literature – specifically examining neurobiological substrates of people who are exposed to multiple adversities and still exhibit adaptive development. The authors truly provide “nuanced and multimodal evidence that maturational trajectories of late-maturing association cortices are associated with changing abilities to adapt to psychosocial stressors during adolescence.” Analyses are sophisticated and creative. Figures are beautiful and help to clearly tell the story of the manuscript. I loved how clearly the Discussion was written and enjoyed reading the thoughtful insights that the authors provided. I have some suggestions below that I feel will improve the quality of the manuscript and I hope that the authors find the feedback helpful. I must admit that I greatly enjoyed reading this manuscript, in fact it may be one of the most well-written + most scientifically informative manuscripts I have read in the last year. Here is my feedback:

We greatly appreciate the kind words and are very happy to learn that you enjoyed our manuscript. Thank you for the constructive comments, which we addressed below.

2.1. For me, it was difficult to understand the exact aims of the study from the abstract and the end of the first paragraph in the Introduction. I was able to figure out the main aims by examining Figure 1 and Figure 2, but it would be helpful if the authors could be a bit more explicit in these two sections.

Thank you for pointing out this lack of clarity. We hope we have now been able to provide a clearer explanation of the aims of the study in the Abstract and Introduction:

We edited and included the following sentences in the abstract:

“Adolescence is a period of dynamic brain remodeling and susceptibility to psychiatric risk factors, mediated by the protracted consolidation of association cortices. Here, we investigated whether longitudinal variation in adolescents’ resilience to psychosocial stressors during this vulnerable period is tied to ongoing myeloarchitectural maturation and consolidation of functional networks. We employed repeated myelin-sensitive Magnetic Transfer (MT) and resting-state functional neuroimaging (n=141) and captured adversity exposure by adverse life events, dysfunctional family settings, and socio-economic status at two timepoints, 1-2yrs apart.”

And edited (among others) the following sentences in the introduction:

For the 2nd paragraph, we now aim to explain our focus on myelin maturation better by starting and ending with the following sentences:

“Insights into adolescent brain development have recently been extended from analyses of cortical size metrics (such as volume and thickness) towards more fine-grained proxies of intra-cortical myelin maturation (Paquola, Bethlehem, et al., 2019; Whitaker et al., 2016; Ziegler et al., 2019). This line of research highlights the continuous myelination of intra- and inter-regional connections, enhancing circuit efficiency as a central feature of adolescent cortical maturation (Mount & Monje, 2017; Paus, 2010).”

[...]

“This protracted maturation, implying longer periods of developmental plasticity, likely reflects later refinement of functional networks associated with abstract cognitive functions, such as cognitive control. However, it also renders them more susceptible to environmental impact and psychopathological alterations (Larsen et al., 2023; Paquola, Bethlehem, et al., 2019; Sydnor et al., 2021b). Thus, the dual role of myelin in structural consolidation and dynamic functional adaptation makes the study of ongoing adolescent myelination a compelling focus to address the question whether the maturation of behavioral capacities for psychosocial adaptation is tied to ongoing cortical consolidation.”

For the final paragraph, we now summarize the main aims of the study, as follows:

“Together, previous research suggests that 1) understanding the development of psychosocial resilience requires complementary longitudinal studies, 2) the protracted consolidation of association cortices by myelination throughout adolescence likely confers increased susceptibility to adverse environmental influences, and 3) in vivo myelin mapping has facilitated multi-modal and multi-scale insights into cortical development. On this basis, the current study investigated whether intra-individual changes in susceptibility and resilience to environmental adversity exposure is tied to differential rates of local and global myeloarchitectural consolidation, and accompanying functional maturation, during adolescence and young adulthood (age range: 14-26yrs).”

2.2. It would also be helpful to have the final sample size (N=141) and number of time points specified in the abstract.

We agree and have added the final sample size as well as the number of time points to the abstract.

“We employed repeated myelin-sensitive Magnetic Transfer (MT) and resting-state functional neuroimaging (n=141) and captured adversity exposure by adverse life events, dysfunctional family settings, and socio-economic status at two timepoints, 1-2yrs apart.”

2.3. It is confusing to have the abbreviation “SRS” used to reflect the stressor resilience score, given that there is a widely used measure of “Social Responsiveness Scale”, a self- and/or parental report that measures autistic traits continuously. I am guessing that the authors chose this abbreviation to reduce their word count. Therefore, I respectfully ask the editor if the authors can increase the word count to account for this increase in words. I found this to be one of the more distracting pieces of the manuscript (constantly re-oriented myself to what SRS stood for in this study).

Thank you for addressing this concern. Indeed, our stressor resilience score (SRS) mirrors the ‘stressor reactivity score’ (SRS) that is more familiar in the resilience literature, e.g., as established by (Kalisch et al., 2021). We agree that this can be confusing. We have now changed the name of the ‘stressor resilience score’ to ‘resilient psychosocial functioning’ score (Re_{PSF}), which is a term that has been used by other groups assessing resilient

psychosocial functioning the way we do (Van Harmelen et al., 2017). We further reduced the number of abbreviations used in the text to improve readability.

2.4. The discussion of how the authors examined how the change in MT – change in social resilience score relationships are linked to “specific cytoarchitectonic cortical types along a histologically defined hierarchy of microstructural complexity” comes out of the blue for me. How did the authors do this? It would be helpful to add a sentence or two here to provide a bit of context. I recognize the methods section is separate, but I was able to understand all of the other methods enough to understand the results, except for this one method.

Thank you for pointing out that our description was too short and unclear. We agree with this point and expanded the description in the results section:

“We next assessed whether effects of longitudinal variation in resilient psychosocial functioning were concentrated in regions characterized by a specific cytoarchitecture, and associated duration of developmental plasticity, using cortical types. The five cortical types comprised agranular, dysgranular, eulaminate I, II and III, and koniocortex and have been proposed to represent a hierarchy of cortical architectonics, ranging from highly differentiated and myelinated koniocortex to less differentiated and more plastic agranular cortex (García-Cabezas et al., 2017, 2019). We stratified the unthresholded t-map according to this prior categorization of cortical types and identified which cortical types overlapped with the significant prefrontal cluster.”

2.5. Like there is a section titled “Fundamental patterns of MT maturation in the full imaging sample”, it would be helpful to see a results section on “changes in social resilient social scores” in the imaging sample. A descriptive characterization of this score and how it changes over time would be helpful for a fuller understanding of the student.

Thank you for this comment. We believe a visualization is the most intuitive way to show changes in resilient psychosocial functioning and we have added panels to **Figure 1A** – bottom right panel, where the scores are introduced (please see bottom panel below). As there is no general age effect on the resilience scores, there is barely a group-average change between timepoints (see lower left panel), which is why we further added individual slopes in the lower right panel.

A | Analysis workflow

B | Intra-individual MT change

Figure 1. Behavioral analysis workflow and group-average longitudinal change in myelin-sensitive Magnetic Transfer (MT). **A**) Resilient psychosocial functioning (RES_{PSF}) scores were computed for each subject at each available time point by predicting psychosocial distress (left) from adversity assessments (Alabama parenting questionnaire (APQ), Life events questionnaire (LEQ), Childhood trauma questionnaire (CTQ), Measure of Parenting style (MOPS), and socio-economic status (SES)). RES_{PSF} scores were defined as the difference between observed and predicted distress, i.e., showing higher (i.e., more susceptible) or lower (i.e., more resilient) than expected psychosocial distress. Longitudinal changes in RES_{PSF} are depicted in the bottom panel. **B**) i) Mean and SD of intra-individual change in myelin-sensitive Magnetic Transfer (Δ MT) in the full imaging sample ($n=199$). ii) Δ MT averaged across the cortex and visualized for three age strata. iii) Mean and SD of Δ MT across 10 intracortical depths and across the cortex. Line colors for ii) and iii) reflect age strata defined in the Middle panel.

We further included a results section titled “Intra-individual changes in resilient psychosocial functioning” alongside a supplementary table (Table S2) where we indicate sample means and potential age and sex effects in both the smaller imaging sub-sample and the larger behavioral sub-sample.

“Intra-individual changes in resilient psychosocial functioning

To study longitudinal variation in resilient psychosocial functioning, we assessed the change (Δ) in Res_{PSF} scores between the first and last measurement timepoint (on average 1.14 (SD 0.32) years apart) in $n = 141$ individuals for whom both repeated imaging and behavioral assessments were available (Figure 1A). 57% of individuals showed a positive change in resilient psychosocial functioning with age (mean $\Delta=2.40$; $SD=16.35$). We did not observe sex differences or age effects on changes in Res_{PSF} scores in either this subsample or the larger behavioral prediction sample in which Res_{PSF} scores were computed ($n=455$ out of $n=712$ individuals with at least two measurement timepoints) (Supplementary Table S2). Longitudinal changes in resilient psychosocial functioning were not related to changes in stressor

exposure (Supplementary Figure S1) and showed comparable distributions in individuals included in imaging analyses and individuals included in behavioral analyses only (Supplementary Figure S2).“

2.6. I became confused when I read this section “....previously established principal axis of age-related change (Figure 3C-E), which captures a differentiation of idiosyncratic sensory vs. paralimbic/temporal maturational patterns”. Could the authors more explicitly describe the exact pattern they are talking about? All I understand from reading this is there is already an age-associated change found, and it differentiates between sensory and temporal regions. However, I don’t have any idea what this pattern looks like and I would like to know that.

Thank you for pointing out this lack of clarity. We believe the confusion may be caused by the broad reference to ‘Figure 3C-E’, where C visualizes how the main axis of age-related change in MPC is conceptually computed, and 3E shows the map that we are referring to in-text and that has previously been established by Paquola and colleagues. We included the alignment with this previously established low-dimensional axis to demonstrate convergence across analytical approaches. We have now referenced the figure more explicitly to make sure readers know what the pattern looks like (Figure 3E) and we have extended the description of the axis in the text:

“The MI_{MPC} spatially aligned with a previously established cortical axis of age-related MPC change (Figure 3C). Regions closer on this axis exhibit more similar patterns of age-related change in MPC, whereas distant regions undergo dissimilar development (Paquola, Bethlehem, et al., 2019). The axis highlights a differentiation of heteromodal frontoparietal cortices that we observed to exhibit disruptive re-organization, to resemble either idiosyncratic sensory or paralimbic/temporal cortex maturational patterns (Figure 3E). At the same time, we observed a U-shaped association between the MI_{MPC} and the principal axis of age effects, where regions at the extremes of the axis exhibited a positive MI_{MPC} , i.e., a positive correlation between baseline and change patterns.”

Reviewer #3

This is an elegant and interesting paper using the NSPN consortia data to understand how changes in myelination in adolescence are related to changes in resilient adaptation. The authors report a positive association between left-lateralised anterolateral prefrontal cortex myelination and more resilient functioning. They then investigated changes in functional connectivity using this region as a seed, more segregation in global connectivity patterns was associated with less resilient functioning.

I think the paper is relatively well-written and addresses an issue that is relatively unaddressed in the field. I found the use of a microstructural covariance matrix to understanding similarities in neural profiles interesting and compelling. My chief concerns are that the methods used are complex and valuable, but not enough has been done to show that they are reliable and meaningful (for example, does the maturational index have value beyond this paper?).

We thank the Reviewer for the positive remarks and constructive comments.

We address the specific comments made by the Reviewer below and would like to take the opportunity to comment on the concern whether the microstructural maturational index (MI_{MPC}) is reliable and meaningful, and whether it has value beyond this paper. Overall, the MI_{MPC} is a direct extension of the previously established maturational index of functional connectivity which captures functional re-organization of association cortices during adolescence and is sensitive to e.g., sex-differences in functional maturation (Dorfschmidt et al., 2022; Váša et al., 2020). It further builds on recent findings suggesting parallel cortical refinement processes at multiple scales, highlighting the relevance of studying synchronized intra-cortical changes in addition to cortical shapes and mean myelin content (Paquola, Bethlehem, et al., 2019; Park et al., 2022; Whitaker et al., 2016; Ziegler et al., 2019). We agree with the importance of demonstrating robustness of the measure and are happy to further clarify the meaningfulness and future utility of this approach.

We have added three paragraphs addressing these points in a supplementary section titled:

“Robustness, estimated maturational processes, and future utility of the MPC maturational index”

1) Is it meaningful?

“The maturational index based on microstructural profile similarity (MI_{MPC}) is a structural extension to the previously established functional connectivity maturational index, capturing conservative (i.e. strengthening of existing connections) and disruptive maturational modes (i.e., a reorganization of existing connectivity patterns; Váša et al., 2020). While the functional maturational index identifies increasing levels of reorganization from unimodal (little reorganization) to transmodal cortex (stronger reorganization), the microstructural maturational index mirrors insights gained from a previously established cortical topology of synchronized age effects on microstructural profile covariance (Paquola, Bethlehem, et al., 2019). In the MI_{MPC} , we observe strongest re-organization in frontoparietal heteromodal cortex, whereas a ‘frame’ of ventral/paralimbic and dorsal/somatosensory cortex shows mostly an age-related strengthening of existing MPC patterns (i.e., little re-organization). This aligns well with what has been reported by Paquola & Bethlehem et al.: association cortical areas, in which overall intra-cortical myelin content increases, develop towards a more ‘sensory-like’ architecture, whereas regions in which preferably mid-to-deeper layers show increases in myelin develop towards a more ‘paralimbic-like’ architecture. This differentiating process is reflected in the ‘disruptive re-organization’ captured by the MI_{MPC} , and mirrors modular segregation observed in tractography-based adolescent data (Baum et al., 2017) and multi-modal assessments of adolescent structural network maturation (Park et al., 2022). At the same time, paralimbic-temporal/ventral and somatosensory/dorsal regions show ‘conservative development’ (i.e., little re-organization) in the MI_{MPC} , suggesting their MPC patterns are well-defined prior to adolescence (Grydeland et al., 2019; Paquola, Bethlehem, et al., 2019). Overall, the MI_{MPC} pattern meaningfully captures synchronized microstructural maturation and re-organization in line with previous observations.”

2) Is it robust?

“To probe the robustness of the MI derived from microstructural data, we repeated the computation of the MI_{MPC} based on different subsamples. First, we drew 100 sub-samples each containing 80% of the individuals per NSPN age bin and repeated the analysis 100 times. The average correlation between the MI_{MPC} map based on all individuals for whom MT data was available ($n = 295$ subjects, 512 sessions/datapoints) and MI_{MPC} maps derived from 80%-sub-samples was $r = 0.96$ (Supplementary Figure S7A). Next, we assessed whether the MI_{MPC} pattern can be observed in smaller sample sizes as well. To this end, we again drew sub-samples per age bin, but this time the size of the subsamples ranged between 20% and 100% (in steps of 5%) of individuals (see Supplementary Figure S7B). Last, results stayed consistent when computing the MI_{MPC} – which reflects the correlation between baseline and age-related change patterns – based on Spearman’s or Pearson’s correlation ($r = 0.99$; Supplementary Figure S7C).”

Supplementary Figure S7. Robustness of the MPC maturational index. A) Histogram depicting correlations between the microstructural profile covariance maturational index (MI_{MPC}) derived from all 295 individuals (512 sessions) and the MI_{MPC} based on 100 sub-samples (80% of data). B) Correlations between the MI_{MPC} derived from all 295 individuals (512 sessions) and the MI_{MPC} based on subsamples of different sizes, ranging from 25-100% in steps of 5%. C) The MI is computed as the correlation between baseline and change patterns. Here, we applied both Spearman’s and Pearson’s correlation to compute the MI and correlated the MI pattern resulting from either method.

3) Can it be used beyond this paper?

“The maturational index based on microstructural profile covariance adds to our understanding of inter-regionally synchronized cortical maturation, capturing adolescent re-organization (integration and segregation) of primarily frontoparietal association cortex. That is, emerging work highlights integrated multi-scale approaches to elucidate biological risk factors associated with neuropsychiatric conditions. It is increasingly recognized that pathological functional perturbations are coupled to microstructural perturbations (Lariviere et al., 2019; Park et al., 2021; Yang et al., 2016; Zheng et al., 2019). Moreover, taking a nuanced approach to studying intracortical myeloarchitectural profiles, beyond mean myelin content, has revealed parallel maturational processes at different scales and topologies (Paquola, Bethlehem, et al., 2019; Whitaker et al., 2016; Ziegler et al., 2019). The use of the MI_{MPC} in future studies may mirror the current use of similar, already established measures such as the main axis of MPC age effects and the maturational index for functional connectivity. That is, the main axis of MPC age effects has already been linked to the cortical topology of microstructural profiles and histology (Paquola, Bethlehem, et al., 2019), which in turn has been combined with other measures of ‘cortical wiring’ in adolescence (Park et al., 2022). Assessing the topology of synchronized structural maturation based on intra-cortical profiles further extends classical structural covariance approaches, assessing e.g. cortical thickness covariance (Raznahan et al., 2011), towards more nuanced myeloarchitecture. Last, similar to our study, the previously established maturational index for functional connectivity (Váša et al., 2020) has been demonstrated to robustly capture sex differences in

adolescent functional network maturation (Dorfschmidt et al., 2022). In sum, the MI_{MPC} may be of interest for future studies addressing adolescent cortical maturation.”

I had the following questions for the authors:

3.1 How do we know the relationship between myelination and psychosocial functioning is specific? Changes in myelination may also be driven by other ongoing changes in adolescence, such as cognitive changes and changes in environments experienced by adolescents, or even third factors such as SES?

Thank you for this comment and the opportunity to explain our interpretation of results further. We agree that biopsychosocial processes are highly interrelated, such that environmental impacts or cognitive changes very likely play into the observed relationship between rates of prefrontal myelination and changes in resilient psychosocial functioning. Notwithstanding, we believe these factors do not contradict our findings and do not necessarily represent confounding factors. In fact, we do not expect maturation of psychosocial functioning to be separate from cognitive maturation, but we would expect cognitive maturation to facilitate (resilient) psychosocial functioning.

The current study takes a first step in demonstrating that adolescent development of resilience capacities is associated with prefrontal myelin maturation. The underlying mechanisms, that is, *why* enhanced myelination is beneficial 1) may be explained by enhanced cognitive strategies at the functional level and enhanced structural stability to stress-induced (e.g., cortisol-level-related) re-modeling at the molecular level. 2) It may be supported by protective environmental factors, which should be investigated as targets in future studies.

1. *Cognitive changes*: Ongoing prefrontal myelination is a characteristic feature of adolescent brain development that optimizes circuit efficiency, which in turn is intricately linked to the maturation of cognitive functions such as executive functions and emotion regulation, and enhanced social and cognitive flexibility required for adaptation (Nelson & Guyer, 2011; Teffer & Semendeferi, 2012). Indeed, enhanced myelination of prefrontal circuits has e.g. been linked to better executive function performance in mice and humans (Maas et al., 2017, 2020; Zhao et al., 2022), and in turn, executive function is a transdiagnostic predictor of mental health symptoms (Etkin et al., 2013). Moreover, several cognitive measures have been shown to be predictive of youth mental health symptoms (Kjelkenes et al., 2023) and resilience (Booth et al., 2022). The tight link between cognitive developmental trajectories and mental health is thus an important focus of behavioral risk and resilience research, and certainly strongly interrelated at the brain level as well. We expect that the question “why” enhanced prefrontal myelination may be beneficial for resilience finds its answer at least partly in enhanced cognitive strategies. For instance, enhanced executive function supported by prefrontal myelination may facilitate coping with/adaptation to environmental stressors by enhanced emotion regulation, maintained cognitive flexibility to find alternative solutions to environmental challenges, and engaging in goal-directed actions even in the face of adversity. We therefore hypothesize cognitive changes to mediate rather than confound the association between prefrontal myelination and changes in resilient psychosocial functioning.
2. *Environmental changes*: Both risk and protective factors may affect neurodevelopmental trajectories and the association we observed with resilience scores.
 - A) Regarding risk factors, we tested whether changes in our behavioral resilient psychosocial functioning (RES_{PSF}) score were the result of changes in risk factors (including various measures of family settings and adverse life events). We did not observe associations between change in RES_{PSF} and changes in adversity exposure (APQ, MOPS, LEQ; Supplementary Figure S1), or changes in adversity measures and changes in cortical myelination (all $p > 0.05$). There were also barely any changes in measured socio-economic status / IMD in included individuals. Other environmental circumstances not measured here may exert influences, which we now address in our discussion of study limitations.
 - B) Regarding protective environmental factors, we agree that beneficial / enriched environments can facilitate cognitive-emotional maturation and may exert their protective effect partly by facilitating brain maturation, including myelination. This would suggest that the neurobiological resilience factor we investigated (microstructural maturation) could be favorably influenced by the environment. This in turn would suggest

that environmental factors that facilitate prefrontal myelin maturation could represent targets for interventions in adolescents exposed to environmental adversity.

We discuss the relationship between resilience-related and cognitive changes in the discussion:

“A beneficial effect of enhanced prefrontal myelination may directly be linked to the optimization of adaptive cognitive strategies facilitating successful navigation in the ever-changing environment. That is, ongoing plasticity of myelination fosters circuit modification and synchronization through a multitude of parallel mechanisms (Xin & Chan, 2020). These may include regulatory influences on axon conductance to optimize the synchronization of spike arrivals (Ford et al., 2015; Kato et al., 2020), neuronal metabolism and excitability (Larson et al., 2018; Xin et al., 2019), and structural plasticity (Wang et al., 2020; Zemmar et al., 2018). In the prefrontal cortex, the optimization of circuit efficiency is intricately linked to the maturation of cognitive functions such as executive functions, including emotion regulation, and enhanced social and cognitive flexibility required for adaptation (Nelson & Guyer, 2011; Teffer & Semendeferi, 2012). Thus, prefrontal maturation may directly facilitate resilient psychosocial functioning by fostering cognitive strategies such as cognitive re-appraisal, self-awareness about potential maladaptive cognitive biases, or decision making/ problem solving to evaluate the impact of adverse experiences and e.g., seek social support. The attainment of beneficial cognitive strategies may therefore mediate the positive association between prefrontal maturation and resilient psychosocial functioning. Conversely, attenuated prefrontal myelination and impaired executive control have been linked to transdiagnostic mental health impairments (Chini & Hanganu-Opatz, 2021; Etkin et al., 2013; Knowles et al., 2022). Schizophrenia rat models further suggest links between interneuron hypomyelination and cognitive inflexibility (Maas et al., 2020).”

And in the study limitations section:

“Next, current findings suggest individual variation in myeloarchitectural maturation as a potential neurobiological resilience factor, influencing adolescent adaptation to environmental risk factors. The exact mechanisms underlying a potential protective effect cannot clearly be elucidated in the current study but may involve increased structural stability to stress-induced re-modelling and enhanced cognitive maturation. Likely, resilient psychosocial development is closely coupled with the attainment of cognitive strategies that facilitate resilient outcomes (Kjelkenes et al., 2023). At the same time, environmental resilience factors such as social support facilitate resilient outcomes (McLaughlin et al., 2020; Reiter et al., 2021; Van Harmelen et al., 2017), and may in part exert their protective effect through an impact on brain maturational trajectories. Resilient outcomes are assumed to rely on a multi-modal and multi-faceted construct, acknowledging the environments we live in, but also other psychological variables beyond clinical symptomatology (such as positive affect, life satisfaction, personality traits). We cannot clearly disentangle the interaction of different intrinsic and extrinsic influences contributing to an individual’s psychological well-being beyond resilience. While we estimated resilience/susceptibility by adjusting psychosocial well-being for adversity exposure - yielding a residualized psychosocial functioning score - we cannot rule out that derived resilience scores somewhat also reflect influences of other genetic and environmental factors, and noise or measurement error (see Supplementary Information).”

And we show changes in adversity measures in Supplementary Figure S1.

Supplementary Figure S1. Changes in adversity exposure between measurement timepoints. The upper row depicts intra-individual changes in adversity measures for which repeated measures were available. The lower row depicts respective correlations in changes in adversity measures with changes in resilient psychosocial functioning (ΔRes_{PSF}), except for IMD which stayed constant in most individuals. APQ = Alabama parenting questionnaire; MOPS = Measure of Parenting Style; LEQ = Life events questionnaire; IMD = Index of mean deprivation.

3.2 In the discussion, the implication is that myelination of the prefrontal cortex drives resilient adaptation. However, changes in myelination might reflect the use of these different strategies. Does the longitudinal data help to disentangle this?

Thanks for this question! We acknowledge that the question of whether changes in myelination reflect the utilization of different strategies is interconnected with the presumed cognitive maturation. This maturation is thought to be facilitated by enhanced prefrontal myelination and may serve as a mediator between myelination and resilience capacities. The longitudinal approach taken specifically addresses the question whether the dynamic nature of resilience/susceptibility is linked to ongoing cortical maturation and re-organization, potentially identifying a more protective set-up. In addition to the fact that our research question is best addressed with a longitudinal design, we consider this choice particularly valuable when studying adaptation to environmental stressors. That is, cross-sectional analyses may more strongly be impacted by inter-individual differences in environmental settings not measured here and which likely stay more constant within an individual (such as family composition or neighborhood). Moreover, this study did not address genetic predispositions. Cross-sectional analyses may be distorted if an individual carrying a genetic predisposition for mental illness shows generally elevated levels of distress compared to another individual with conceptually comparable levels of environmental stressor exposure. In our longitudinal approach, such potential general offsets in resilience scores due to genetic or other environmental influences are more likely to be present across measurement timepoints than they are to be consistent across individuals, such that assessing intra-individual change would mitigate their confounding effects. We thus believe it indeed allows us to better disentangle cortical maturation and potential psychological and cognitive development from confounding inter-individual factors.

We now briefly discuss the benefits of our longitudinal approach in the discussion:

“Taking a longitudinal approach allowed us to limit confounding effects of individual differences in genetic predispositions or environmental circumstances (such as family composition or neighborhood)

that are likely to contribute less variability to within-subject repeated measures than to between-subject cross-sectional data.”

We further agree that enhanced myelination of prefrontal circuits contributes to the maturation of several executive functions that may in turn increase the likelihood of accessing effective strategies to cope with stressors. This notion is supported by the fact that overall myelin content generally increases across the cortex throughout adolescence, but only prefrontal myelination was implicated in the development of resilience in the current study. This points towards a ‘special role’ of the abstract cognitive processes that the PFC is involved in, including the selection of strategies to respond to the environment. These could include top-down emotion regulation strategies such as positive reappraisal, decision-making facilitating e.g., seeking social support, critical thinking and problem-solving to evaluate the impact of adverse experiences – or developing meta-cognitive skills to foster self-awareness about potential maladaptive cognitive biases. All of these comprise cognitive strategies with the potential to beneficially influence how adolescent well-being is impacted by environmental stressors, and the maturation of which may be enhanced by prefrontal myelination. We would thus interpret the use of different (cognitive) strategies as one variable mediating the effect of myelin maturation on behavioral adaptation - next to physiological aspects such as structural stability limiting the effects of stress-induced network re-modeling. As such, we see it as a likely explanation of our findings rather than a confounding factor. At this point, we can only draw a theoretical link based on existing literature on the role of the prefrontal cortex and cognition for resilience (Eaton et al., 2022; Larsen & Luna, 2018; Luciana & Collins, 2022), as the direct assessment of cognitive strategies was beyond the scope of the current study.

We further hypothesize that other mechanisms beyond changes in cognitive strategies likely underly resilient psychosocial maturation. At the physiological level, enhanced myelination further provides stability by consolidating established connections. This structural stability may protect against glucocorticoid-induced remodeling affecting the prefrontal cortex in response to stress (McEwen, 2007; McEwen et al., 2016).

We have addressed potential cognitive strategies:

“A beneficial effect of enhanced prefrontal myelination may directly be linked to the optimization of adaptive cognitive strategies facilitating successful navigation in the ever-changing environment. That is, ongoing plasticity of myelination fosters circuit modification and synchronization through a multitude of parallel mechanisms (Xin & Chan, 2020). These may include regulatory influences on axon conductance to optimize the synchronization of spike arrivals (Ford et al., 2015; Kato et al., 2020), neuronal metabolism and excitability (Larson et al., 2018; Xin et al., 2019), and structural plasticity (Wang et al., 2020; Zemmar et al., 2018). In the prefrontal cortex, the optimization of circuit efficiency is intricately linked to the maturation of cognitive functions such as executive functions, including emotion regulation, and enhanced social and cognitive flexibility required for adaptation (Nelson & Guyer, 2011; Teffer & Semendeferi, 2012). Thus, prefrontal maturation may directly facilitate resilient psychosocial functioning by fostering cognitive strategies such as cognitive re-appraisal, self-awareness about potential maladaptive cognitive biases, or decision making/ problem solving to evaluate the impact of adverse experiences and e.g. seek social support. The attainment of beneficial cognitive strategies may therefore mediate the positive association between prefrontal maturation and resilient psychosocial functioning. Conversely, attenuated prefrontal myelination and impaired executive control have been linked to transdiagnostic mental health impairments (Chini & Hanganu-Opatz, 2021; Etkin et al., 2013; Knowles et al., 2022). Schizophrenia rat models further suggest links between interneuron hypomyelination and cognitive inflexibility (Maas et al., 2020).”

3.3 The number of children with longitudinal MRI data is relatively small (N=141). What size of effects were expected? Was a power analysis conducted prior to data analysis?

Thank you for expressing your concerns about power – we are happy to further clarify. We have not formally conducted an a-priori power analysis to inform our sample size given that we aimed to include the maximum possible N for which a list of requirements was met (including high-quality, repeated and multi-modal neuroimaging, as well as broad assessments of risk exposure and psychological questionnaires at two time points in a dataset that covers an age range associated with elevated psychiatric vulnerability, namely adolescence). Therefore, the trade-off between ‘deep phenotyping’ vs big datasets led pragmatically to the inclusion of 141 individuals.

However, to further address this concern, we have additionally conducted an exemplary post-hoc power analysis for the analysis linking change in mean myelin / MT to changes in resilient psychosocial functioning. We used the SIMR package in R for power analysis of generalized linear (mixed) models by simulation (Green & MacLeod, 2016) running 200 simulations. Specifying a standardized beta of 0.25, which was the smallest observed effect in the prefrontal cortex, indicated 89% power (CI = [83.3 – 92.9]) to observe this effect for $\alpha < 0.05$. At the same time, we are cautious about formally including a post-hoc power analyses in our manuscript as there is concern over the conceptual meaningfulness of post-hoc power analyses in general, and especially based on observed effects (Quach et al., 2022). Therefore, we refrained from including post-hoc power-analyses, but we have transparently reported that we did not conduct an a-priori power-analysis, and we have pointed out concerns related to the current sample size in the discussion. We further argue that, though based on a comparatively small sample, our presented results are robust to multiple analytical choices and sub-sampling, and align well with existing literature.

We added the following section to the discussion:

“We further acknowledge that the included sample is of respectable but not massive size. The current sample size was not determined by an a-priori power-analysis. Rather, it resulted from the inclusion of a sample for which adolescent, longitudinal, and multi-modal imaging including a myelin proxy, as well as in-depth phenotypic characterization was available. Overall, we observed that present results were robust to model parameter manipulation and sub-sampling, and well in line with the existing literature, highlighting the central role of the PFC in stress adaptivity and vulnerability (Eaton et al., 2022; Luciana & Collins, 2022), as well as the role of association cortex maturation in psychiatric susceptibility (Larsen et al., 2023; Paus et al., 2008; Sydnor et al., 2021a). Moreover, individual differences in rates myelination were observed in regions generally showing highest rates of myelination during adolescence (Figure 1B). This supports a link to the protracted critical period plasticity of the prefrontal cortex well into early adulthood associated with increased susceptibility to environmental risk factors (Larsen et al., 2023). We further observed convergence across modalities, such that, for instance, effects on prefrontal functional network maturation strengthened the meaningfulness of observed differences in rates of prefrontal myelination. However, recent concerns about replicability of neuroimaging studies linked to insufficient sample sizes (Marek et al., 2022) warn about inflated effect sizes. Therefore, we stress that current results should be interpreted with caution, pending replication in future studies including independent and larger samples.”

3.4 Why did the authors choose to focus on MTsat from the MPM data, rather than other measures such as R1 or R2*?

We agree that all MPM parameters reflect characteristic aspects of tissue micro-architecture and thus may have the potential to detect maturational patterns.

MT is considered a particularly strong *in vivo* marker of myelin with a high spatial correspondence with myelin basic protein and other myelin-related molecules in the brain, as has been verified by several histological validation studies (Mancini et al., 2020; Odrobina et al., 2005; Paquola & Hong, 2023; Schmierer et al., 2007). A recent study suggests that amongst MPM parameters, MT and PD showed highest reliability, appeared to carry

more subject-specific information and were thus concluded to be more suitable to study individual differences in brain-behavior associations than R1 and R2* (Wenger et al., 2022). R2* was e.g., found to show significant day-specific variances in regions including the MTG and precuneus, and partly the OFC, suggesting a more limited suitability for individual differences study. Moreover, while being sensitive to myelin as well, the R2* signal is strongly influenced by iron content (Stüber et al., 2014). In the cortex, particularly blood vessels influence R2* gradient-recalled echo signal decay (Weiskopf et al., 2021).

R1/T1 images have increasingly been used in combination with T2 weighted images to assess cortical myelin content, by assessing the T1w/T2w ratio (Glasser & Essen, 2011). One reason why the T1w/T2w ratio has become a popular myelin proxy is that the sequences are more readily available and part of routine scans. However, both T1w and T2w images individually have low specificity for myelin (Paquola & Hong, 2023).

In sum, while R1 and R2* images can provide information about tissue properties beyond myelin content, such as iron concentration, which could have been combined in a multi-modal quantification approach (Draganski et al., 2011), our research question specifically targeted myelin maturation. That is, we selected the NSPN dataset, amongst other reasons, because of the availability of MT data which has been shown to be a valuable myelin proxy with high sensitivity and specificity to myelin content. Here, we add to the existing literature basis demonstrating MT's suitability to detect longitudinal change (Leutritz et al., 2020) involving developmental processes (Paquola, Bethlehem, et al., 2019; Whitaker et al., 2016; Ziegler et al., 2019), and individual differences including pathological alterations in myelin content (Schmierer et al., 2007; Ziegler et al., 2019).

3.5 Did the authors assess the effect of movement on their MPM data in this developmental group? What quality control pipeline was used?

We agree that motion artifacts pose a challenge to neuroimaging analyses in developmental samples. For anatomical data, we accounted for motion effects by strictly excluding scans with motion artifacts rather than taking attempts to remove motion-related effects to retain images. R1/T1 and MT images were visually inspected for motion artifacts (such as ringing, ghosting, smearing or blurring) by experts and scans for which motion artifacts were present were excluded from analyses. Moreover, six independent members of the NSPN Consortium carried out the quality control of reconstructed surfaces. Surface reconstructions / segmentations were edited by adding control points in FreeSurfer, re-processed, and then underwent quality control again. If further motion artifacts were detected in this process, the relevant scans were excluded.

Overall, 17 scans were excluded due to high in-scanner motion in either anatomical (3 participants) or functional data (here defined as mean FD > 0.3 mm or maximum FD > 1.3 mm).

Thank you for pointing out that this QC step was not mentioned in our methods section – we now added:

“T1w and MT images were visually inspected for motion artifacts (such as ringing, ghosting, smearing or blurring) by experts and scans were strictly excluded if motion artifacts were detected.”

[...]

“Surface reconstructions / segmentations were edited by adding control points in FreeSurfer, re-processed, and then underwent quality control again. If further motion artifacts were detected in this process, the relevant scans were excluded.”

3.6 How reliable was the prediction of distress scores from adversity measures?

We are happy to elaborate on this point. For our prediction of distress scores from The Life Events Questionnaire, the Child Trauma Questionnaire, the Alabama Parenting Questionnaire, the Measure of Parenting Style, and socioeconomic status, we observed an R² of 0.21 (MAE = 15.15, correlation between true and predicted distress scores: r = 0.46). The amount of variance explained by our model (approx. 21%) is comparable with previous work reporting 24% of variance explained by the prediction of psychosocial functioning from family experiences in longitudinal settings (Van Harmelen et al., 2017), 21% when predicting internalizing symptoms from general life stressors during the covid pandemic (Veer et al., 2021), or 28% when predicting psychosocial functioning from childhood adversity (González-García et al., 2023).

We made several analytical choices to increase robustness of the model and by which we adapted but partly diverged from previously established regression models to derive resilience scores (e.g. (Collishaw et al., 2016; Kalisch et al., 2021; Miller-Lewis et al., 2013; Sapouna & Wolke, 2013; Van Harmelen et al., 2017)). In contrast to common resilience models, we implemented a nested cross-validation including hyper-parameter tuning (number of estimators and tree depth) to prevent overfitting and improve reliability of the model. Moreover, we used a random forest regression allowing us to find a non-parametric and non-linear solution to account for the frequently observed skewness in risk variables (e.g., the number of individuals with severe childhood trauma is typically lower than the number of individuals with little childhood trauma). It further diminished the need to compare linear, quadratic, cubic etc. solutions and enhanced parsimony. Given that we were specifically interested in intra-individual change and thus expected our outcome measure to be unstable between timepoints, we considered the assessment of test-retest reliability of the longitudinal prediction via e.g. ICC not meaningful in this case.

We included a brief discussion on this matter in the limitations & future directions section:

“Resilient outcomes are assumed to rely on a multi-modal and multi-faceted construct, acknowledging the environments we live in, but also other psychological variables beyond clinical symptomatology (such as positive affect, life satisfaction, personality traits). We cannot clearly disentangle the interaction of different intrinsic and extrinsic influences contributing to an individual’s psychological well-being beyond resilience. While we estimated resilience/susceptibility by adjusting psychosocial well-being for adversity exposure - yielding a residualized psychosocial functioning score - we cannot rule out that derived resilience scores somewhat also reflect influences of other genetic and environmental factors, and noise or measurement error (see Supplementary Information). For instance, potential self-report / retrospectivity biases inherent to measures of well-being and adversity exposure (Baldwin et al., 2019) would persist in resilience scores, but are not caused by the residual approach providing resilience scores. Taking a longitudinal approach allowed us to limit confounding effects of individual differences in genetic predispositions or environmental circumstances (such as family composition or neighborhood) that are likely to contribute less variability to within-subject repeated measures than to between-subject cross-sectional data. Overall, our model (explaining 21% of variance) controls for exposure to a comparable degree as common resilience models (explaining 21-28% of variance; see e.g., (González-García et al., 2023; Van Harmelen et al., 2017; Veer et al., 2021)). We further aimed to increase robustness and generalizability of the model by implementing a nested cross-validation and a random forest regression robust to non-linear and non-parametric distributions of questionnaire data.”

3.7 How easy was the alignment and registration between the MPM data and the functional data? How was this checked?

We are happy to further clarify. Functional data was registered to T1/R1 images, and MT and T1/R1 images are MPM modalities acquired in the same sequence, which makes them well-aligned by construction. Thus, alignment parameters determined to align functional to R1/T1 data can be applied to other MPM modalities including MT.

Co-registration of functional and T1/R1 images was done as part of functional preprocessing using AFNI’s multi-echo independent component analysis (ME-ICA). Volumes obtained within the 15-second steady-state equilibration were excluded. Anatomical-functional co-registration and motion correction parameters were computed using the middle TE data, and the base EPI image was the first volume following equilibration. Matrices for de-obliquing and six-parameter rigid body motion correction were computed. Using the LPC cost function with the EPI base image as the LPC weight mask, a 12-parameter affine anatomical-to-functional co-registration was computed. Matrices for de-obliquing, motion correction, and anatomical-functional co-registration were combined into a single alignment matrix using the concatenation approach from the AFNI tool *align_epi_anat.py*. The dataset of each TE was then slice-time corrected and spatially aligned through application of the alignment matrix.

Moreover, MT and functional data were parcellated into the same 360 cortical regions using the HCP parcellation (Glasser et al., 2016). The parcellation was defined on the FreeSurfer standard anatomical template

(fsaverage), and subsequently transformed to each individual subject's surface (i.e., in native space). Each subject's surface parcellation was interpolated and expanded to their respective R1 and MT images. For functional data, surface parcellations were transformed into volume space using Freesurfer command `mri_aparc2aseg`, allowing us then to extract and average regional BOLD time series over all voxels in each volumetric parcel.

Registrations were visually inspected, and 9 subjects were excluded due to co-registration errors.

We added this information to the Methods section and thank the Reviewer for pointing out this lack of clarity:

“Functional data were co-registered to R1 images, which were derived from the same MPM sequence as MT data, ensuring spatial alignment between functional and MT data. Volumes obtained within the 15-second steady-state equilibration were excluded. Anatomical-functional co-registration and motion correction parameters were computed using the middle TE data, and the base EPI image was the first volume following equilibration. Matrices for de-obliquing and six-parameter rigid body motion correction were computed. Using the LPC cost function with the EPI base image as the LPC weight mask, a 12-parameter affine anatomical-functional co-registration was computed. Matrices for de-obliquing, motion correction, and anatomical-functional co-registration were concatenated into a single alignment matrix using the AFNI tool `align_epi_anat.py`. The dataset of each TE was then slice-time corrected and spatially aligned through repeated application of the alignment matrix.”

3.8 Although the microstructural covariance matrix argument is compelling, from the paper, it is really hard to understand how this is being calculated, and therefore evaluate further arguments. It would be helpful if this was clearer.

Thank you for pointing out that the computation of microstructural profile covariance (MPC) matrices was unclear: we are happy to further clarify. The MPC matrix is designed to capture the similarity of intra-cortical MT intensity profiles between regions. An underlying assumption of the MPC approach is that cortical regions that show higher microstructural similarity are more likely anatomically connected (García-Cabezas et al., 2019). While microstructural profiles generated here do not have the resolution to assess similarities in cytoarchitectonic profiles, they yield information about estimated myelin content at different cortical depths, and how this distribution of intra-cortical myelin content shows similarities and differences between regions. We can then assess age effects on the similarity between the profiles of any two regions. A positive age effect indicates that the distributions of intracortical myelin content of two regions become more similar to each other with age, likely reflecting a structural integration of these two regions (García-Cabezas et al., 2019). By assessing age effects on all edges of the MPC matrix (sometimes referred to as the MPC network (Paquola, Bethlehem, et al., 2019)), we can study the cortical topology of synchronized developmental patterns and longitudinal re-organization of the MPC network.

Two main steps are required to compute a MPC matrix: 1) The generation of an intra-cortical profile for each region. These profiles are generated by extracting MT intensity values at 10 intra-cortical depths (approximating variance in myelin content along a radial transect from the cortical surface to the white matter boundary). And 2) Computing the pairwise Pearson correlation between regional MT profiles, controlling for the average MT intensity across cortical depth.

We acknowledge that this explanation is already needed in the results section to help the reader understand our system-level analyses and we have therefore added the following sentences to the results section:

“Thus, we next aimed to study system-level myeloarchitectural and parallel functional re-organization. To this end, we computed a microstructural profile covariance (MPC) network reflecting inter-regional similarities of myeloarchitectural profiles. The MPC matrix was generated by first probing MT intensities at ten equally spaced intra-cortical depth coordinates, yielding cortical depth profiles of regional MT from the pial surface to the white matter boundary of each cortical area. We then calculated the pairwise Pearson correlation between regional profiles while controlling for average MT intensity across cortical depth to derive the MPC matrix (Figure 3A). This allowed us to study the topology of

synchronized effects of age on depth-specific changes in approximated myelin content, which are reflected in changes in regional intra-cortical profiles and thus their inter-regional similarities.

We further added a visualization of the MPC matrix generation to Figure 3A:

A | Microstructural profile covariance

References

- Baldwin, J. R., Reuben, A., Newbury, J. B., & Danese, A. (2019). Agreement Between Prospective and Retrospective Measures of Childhood Maltreatment: A Systematic Review and Meta-analysis. *JAMA Psychiatry*, *76*(6), 584–593. <https://doi.org/10.1001/jamapsychiatry.2019.0097>
- Baum, G. L., Ciric, R., Roalf, D. R., Betzel, R. F., Moore, T. M., Shinohara, R. T., Kahn, A. E., Vandekar, S. N., Rupert, P. E., Quarmley, M., Cook, P. A., Elliott, M. A., Ruparel, K., Gur, R. E., Gur, R. C., Bassett, D. S., & Satterthwaite, T. D. (2017). Modular Segregation of Structural Brain Networks Supports the Development of Executive Function in Youth. *Current Biology*, *27*(11), 1561–1572.e8. <https://doi.org/10.1016/j.cub.2017.04.051>
- Booth, C., Songco, A., Parsons, S., & Fox, E. (2022). Cognitive mechanisms predicting resilient functioning in adolescence: Evidence from the CogBIAS longitudinal study. *Development and Psychopathology*, *34*(1), 345–353. <https://doi.org/10.1017/S0954579420000668>
- Bowes, L., Maughan, B., Caspi, A., Moffitt, T. E., & Arseneault, L. (2010). Families promote emotional and behavioural resilience to bullying: Evidence of an environmental effect. *Journal of Child Psychology and Psychiatry, and Allied Disciplines*, *51*(7), 809–817. <https://doi.org/10.1111/j.1469-7610.2010.02216.x>
- Buckley, L., Broadley, M., & Cascio, C. N. (2019). Socio-economic status and the developing brain in adolescence: A systematic review. *Child Neuropsychology*, *25*(7), 859–884. <https://doi.org/10.1080/09297049.2018.1549209>
- Chini, M., & Hanganu-Opatz, I. L. (2021). Prefrontal Cortex Development in Health and Disease: Lessons from Rodents and Humans. *Trends in Neurosciences*, *44*(3), 227–240. <https://doi.org/10.1016/j.tins.2020.10.017>
- Collishaw, S., Hammerton, G., Mahedy, L., Sellers, R., Owen, M. J., Craddock, N., Thapar, A. K., Harold, G. T., Rice, F., & Thapar, A. (2016). Mental health resilience in the adolescent offspring of parents with depression: A prospective longitudinal study. *The Lancet. Psychiatry*, *3*(1), 49–57. [https://doi.org/10.1016/S2215-0366\(15\)00358-2](https://doi.org/10.1016/S2215-0366(15)00358-2)
- Dorfschmidt, L., Bethlehem, R. A., Seidlitz, J., Váša, F., White, S. R., Romero-García, R., Kitzbichler, M. G., Aruldass, A. R., Morgan, S. E., Goodyer, I. M., Fonagy, P., Jones, P. B., Dolan, R. J., NSPN Consortium, Harrison, N. A., Vértes, P. E., & Bullmore, E. T. (2022). Sexually divergent development of depression-related brain networks during healthy human adolescence. *Science Advances*, *8*(21), eabm7825. <https://doi.org/10.1126/sciadv.abm7825>
- Draganski, B., Ashburner, J., Hutton, C., Kherif, F., Frackowiak, R. S. J., Helms, G., & Weiskopf, N. (2011). Regional specificity of MRI contrast parameter changes in normal ageing revealed by voxel-based quantification (VBQ). *NeuroImage*, *55*(4), 1423–1434. <https://doi.org/10.1016/j.neuroimage.2011.01.052>
- Eaton, S., Cornwell, H., Hamilton-Giachritsis, C., & Fairchild, G. (2022). Resilience and young people’s brain structure, function and connectivity: A systematic review. *Neuroscience & Biobehavioral Reviews*, *132*, 936–956. <https://doi.org/10.1016/j.neubiorev.2021.11.001>
- Etkin, A., Gyurak, A., & O’Hara, R. (2013). A neurobiological approach to the cognitive deficits of psychiatric disorders. *Dialogues in clinical neuroscience*, *15*(4), Article 4.
- Ford, M. C., Alexandrova, O., Cossell, L., Stange-Marten, A., Sinclair, J., Kopp-Scheinflug, C., Pecka, M., Attwell, D., & Grothe, B. (2015). Tuning of Ranvier node and internode properties in myelinated axons to adjust action potential timing. *Nature Communications*, *6*(1), Article 1. <https://doi.org/10.1038/ncomms9073>
- García-Cabezas, M. Á., Joyce, M. K. P., John, Y. J., Zikopoulos, B., & Barbas, H. (2017). Mirror trends of plasticity and stability indicators in primate prefrontal cortex. *European Journal of Neuroscience*, *46*(8), 2392–2405. <https://doi.org/10.1111/ejn.13706>
- García-Cabezas, M. Á., Zikopoulos, B., & Barbas, H. (2019). The Structural Model: A theory

linking connections, plasticity, pathology, development and evolution of the cerebral cortex. *Brain Structure and Function*, 224(3), 985–1008. <https://doi.org/10.1007/s00429-019-01841-9>

Garcini, L. M., Arredondo, M. M., Berry, O., Church, J. A., Fryberg, S., Thomason, M. E., & McLaughlin, K. A. (2022). Increasing diversity in developmental cognitive neuroscience: A roadmap for increasing representation in pediatric neuroimaging research. *Developmental Cognitive Neuroscience*, 58, 101167. <https://doi.org/10.1016/j.dcn.2022.101167>

Glasser, M. F., Coalson, T. S., Robinson, E. C., Hacker, C. D., Harwell, J., Yacoub, E., Ugurbil, K., Andersson, J., Beckmann, C. F., Jenkinson, M., Smith, S. M., & Van Essen, D. C. (2016). A multi-modal parcellation of human cerebral cortex. *Nature*, 536(7615), 171–178. <https://doi.org/10.1038/nature18933>

Glasser, M. F., & Essen, D. C. V. (2011). Mapping Human Cortical Areas In Vivo Based on Myelin Content as Revealed by T1- and T2-Weighted MRI. *Journal of Neuroscience*, 31(32), 11597–11616. <https://doi.org/10.1523/JNEUROSCI.2180-11.2011>

González-García, N., Buimer, E. E. L., Moreno-López, L., Sallie, S. N., Váša, F., Lim, S., Romero-Garcia, R., Scheuplein, M., Whitaker, K. J., Jones, P. B., Dolan, R. J., Consortium, N., Fonagy, P., Goodyer, I., Bullmore, E. T., & Harmelen, A.-L. van. (2023). Resilient functioning is associated with altered structural brain network topology in adolescents exposed to childhood adversity. *Development and Psychopathology*, 1–11. <https://doi.org/10.1017/S0954579423000901>

Green, P., & MacLeod, C. J. (2016). SIMR: An R package for power analysis of generalized linear mixed models by simulation. *Methods in Ecology and Evolution*, 7(4), 493–498. <https://doi.org/10.1111/2041-210X.12504>

Grydeland, H., Vértes, P. E., Váša, F., Romero-Garcia, R., Whitaker, K., Alexander-Bloch, A. F., Bjørnerud, A., Patel, A. X., Sederevicius, D., Tamnes, C. K., Westlye, L. T., White, S. R., Walhovd, K. B., Fjell, A. M., & Bullmore, E. T. (2019). Waves of Maturation and Senescence in Micro-structural MRI Markers of Human Cortical Myelination over the Lifespan. *Cerebral Cortex (New York, N.Y.: 1991)*, 29(3), 1369–1381. <https://doi.org/10.1093/cercor/bhy330>

Kalisch, R., Baker, D. G., Basten, U., Boks, M. P., Bonanno, G. A., Brummelman, E., Chmitorz, A., Fernández, G., Fiebach, C. J., Galatzer-Levy, I., Geuze, E., Groppa, S., Helmreich, I., Hendler, T., Hermans, E. J., Jovanovic, T., Kubiak, T., Lieb, K., Lutz, B., ... Kleim, B. (2017). The resilience framework as a strategy to combat stress-related disorders. *Nature Human Behaviour*, 1(11), 784–790. <https://doi.org/10.1038/s41562-017-0200-8>

Kalisch, R., Köber, G., Binder, H., Ahrens, K. F., Basten, U., Chmitorz, A., Choi, K. W., Fiebach, C. J., Goldbach, N., Neumann, R. J., Kampa, M., Kollmann, B., Lieb, K., Plichta, M. M., Reif, A., Schick, A., Sebastian, A., Walter, H., Wessa, M., ... Engen, H. (2021). The Frequent Stressor and Mental Health Monitoring-Paradigm: A Proposal for the Operationalization and Measurement of Resilience and the Identification of Resilience Processes in Longitudinal Observational Studies. *Frontiers in Psychology*, 12. <https://www.frontiersin.org/articles/10.3389/fpsyg.2021.710493>

Kato, D., Wake, H., Lee, P. R., Tachibana, Y., Ono, R., Sugio, S., Tsuji, Y., Tanaka, Y. H., Tanaka, Y. R., Masamizu, Y., Hira, R., Moorhouse, A. J., Tamamaki, N., Ikenaka, K., Matsukawa, N., Fields, R. D., Nabekura, J., & Matsuzaki, M. (2020). Motor learning requires myelination to reduce asynchrony and spontaneity in neural activity. *Glia*, 68(1), 193–210. <https://doi.org/10.1002/glia.23713>

Kjelkenes, R., Wolfers, T., Alnæs, D., van der Meer, D., Pedersen, M. L., Dahl, A., Voldsbakk, I., Moberget, T., Tamnes, C. K., Andreassen, O. A., Marquand, A. F., & Westlye, L. T. (2023). Mapping Normative Trajectories of Cognitive Function and Its Relation to Psychopathology Symptoms and Genetic Risk in Youth. *Biological Psychiatry Global Open Science*, 3(2), 255–263. <https://doi.org/10.1016/j.bpsgos.2022.01.007>

Knowles, J. K., Batra, A., Xu, H., & Monje, M. (2022). Adaptive and maladaptive

myelination in health and disease. *Nature Reviews Neurology*, 18(12), Article 12. <https://doi.org/10.1038/s41582-022-00737-3>

Lariviere, S., Vos de Wael, R., Paquola, C., Hong, S. J., Misic, B., Bernasconi, N., Bernasconi, A., Bonilha, L., & Bernhardt, B. C. (2019). Microstructure-Informed Connectomics: Enriching Large-Scale Descriptions of Healthy and Diseased Brains. *Brain Connect*, 9(2), Article 2. <https://doi.org/10.1089/brain.2018.0587>

Larsen, B., & Luna, B. (2018). Adolescence as a neurobiological critical period for the development of higher-order cognition. *Neuroscience & Biobehavioral Reviews*, 94, 179–195. <https://doi.org/10.1016/j.neubiorev.2018.09.005>

Larsen, B., Sydnor, V. J., Keller, A. S., Yeo, B. T. T., & Satterthwaite, T. D. (2023). A critical period plasticity framework for the sensorimotor–association axis of cortical neurodevelopment. *Trends in Neurosciences*, 0(0). <https://doi.org/10.1016/j.tins.2023.07.007>

Larson, V. A., Mironova, Y., Vanderpool, K. G., Waisman, A., Rash, J. E., Agarwal, A., & Bergles, D. E. (2018). Oligodendrocytes control potassium accumulation in white matter and seizure susceptibility. *eLife*, 7, e34829. <https://doi.org/10.7554/eLife.34829>

Leutritz, T., Seif, M., Helms, G., Samson, R. S., Curt, A., Freund, P., & Weiskopf, N. (2020). Multiparameter mapping of relaxation (R1, R2*), proton density and magnetization transfer saturation at 3 T: A multicenter dual-vendor reproducibility and repeatability study. *Human Brain Mapping*, 41(15), 4232–4247. <https://doi.org/10.1002/hbm.25122>

Luciana, M., & Collins, P. F. (o. J.). *Neuroplasticity, the Prefrontal Cortex, and Psychopathology-Related Deviations in Cognitive Control*.

Maas, D. A., Eijnsink, V. D., Spoelder, M., van Hulst, J. A., De Weerd, P., Homberg, J. R., Vallès, A., Nait-Oumesmar, B., & Martens, G. J. M. (2020). Interneuron hypomyelination is associated with cognitive inflexibility in a rat model of schizophrenia. *Nature Communications*, 11(1), Article 1. <https://doi.org/10.1038/s41467-020-16218-4>

Maas, D. A., Vallès, A., & Martens, G. J. M. (2017). Oxidative stress, prefrontal cortex hypomyelination and cognitive symptoms in schizophrenia. *Translational Psychiatry*, 7(7), e1171–e1171. <https://doi.org/10.1038/tp.2017.138>

Mancini, M., Karakuzu, A., Cohen-Adad, J., Cercignani, M., Nichols, T. E., & Stikov, N. (2020). An interactive meta-analysis of MRI biomarkers of myelin. *eLife*, 9, e61523. <https://doi.org/10.7554/eLife.61523>

Marek, S., Tervo-Clemmens, B., Calabro, F. J., Montez, D. F., Kay, B. P., Hatoum, A. S., Donohue, M. R., Foran, W., Miller, R. L., Hendrickson, T. J., Malone, S. M., Kandala, S., Feczko, E., Miranda-Dominguez, O., Graham, A. M., Earl, E. A., Perrone, A. J., Cordova, M., Doyle, O., ... Dosenbach, N. U. F. (2022). Reproducible brain-wide association studies require thousands of individuals. *Nature*, 603(7902), 654–660. <https://doi.org/10.1038/s41586-022-04492-9>

McEwen, B. S. (2007). Physiology and Neurobiology of Stress and Adaptation: Central Role of the Brain. *Physiological Reviews*. <https://doi.org/10.1152/physrev.00041.2006>

McEwen, B. S., Nasca, C., & Gray, J. D. (2016). Stress Effects on Neuronal Structure: Hippocampus, Amygdala, and Prefrontal Cortex. *Neuropsychopharmacology*, 41(1), Article 1. <https://doi.org/10.1038/npp.2015.171>

McLaughlin, K. A., Colich, N. L., Rodman, A. M., & Weissman, D. G. (2020). Mechanisms linking childhood trauma exposure and psychopathology: A transdiagnostic model of risk and resilience. *BMC Medicine*, 18(1), Article 1. <https://doi.org/10.1186/s12916-020-01561-6>

Miller-Lewis, L. R., Searle, A. K., Sawyer, M. G., Baghurst, P. A., & Hedley, D. (2013). Resource factors for mental health resilience in early childhood: An analysis with multiple methodologies. *Child and Adolescent Psychiatry and Mental Health*, 7(1), 6. <https://doi.org/10.1186/1753-2000-7-6>

Mount, C. W., & Monje, M. (2017). Wrapped to Adapt: Experience-Dependent Myelination. *Neuron*, 95(4), 743–756. <https://doi.org/10.1016/j.neuron.2017.07.009>

Nelson, E. E., & Guyer, A. E. (2011). The development of the ventral prefrontal cortex and social flexibility. *Developmental Cognitive Neuroscience*, *1*(3), 233–245. <https://doi.org/10.1016/j.dcn.2011.01.002>

Odrobina, E. E., Lam, T. Y. J., Pun, T., Midha, R., & Stanis, G. J. (2005). MR properties of excised neural tissue following experimentally induced demyelination. *NMR in Biomedicine*, *18*(5), 277–284. <https://doi.org/10.1002/nbm.951>

Paquola, C., Bethlehem, R. A., Seidlitz, J., Wagstyl, K., Romero-Garcia, R., Whitaker, K. J., Vos De Wael, R., Williams, G. B., Vértes, P. E., Margulies, D. S., Bernhardt, B., & Bullmore, E. T. (2019). Shifts in myeloarchitecture characterise adolescent development of cortical gradients. *eLife*, *8*. <https://doi.org/10.7554/elife.50482>

Paquola, C., & Hong, S.-J. (2023). The Potential of Myelin-Sensitive Imaging: Redefining Spatiotemporal Patterns of Myeloarchitecture. *Biological Psychiatry*, *93*(5), 442–454. <https://doi.org/10.1016/j.biopsych.2022.08.031>

Paquola, C., Vos de Wael, R. V., Wagstyl, K., Bethlehem, R. A., Hong, S.-J., Seidlitz, J., Bullmore, E. T., Evans, A. C., Misic, B., & Margulies, D. S. (2019). Microstructural and functional gradients are increasingly dissociated in transmodal cortices. *PLOS Biology*, *17*(5), Article 5.

Park, B., Paquola, C., Bethlehem, R. A. I., Benkarim, O., Neuroscience in Psychiatry Network (NSPN) Consortium, Mišić, B., Smallwood, J., Bullmore, E. T., & Bernhardt, B. C. (2022). Adolescent development of multiscale structural wiring and functional interactions in the human connectome. *Proceedings of the National Academy of Sciences*, *119*(27), e2116673119. <https://doi.org/10.1073/pnas.2116673119>

Park, B.-Y., Hong, S.-J., Valk, S. L., Paquola, C., Benkarim, O., Bethlehem, R. A. I., Di Martino, A., Milham, M. P., Gozzi, A., Yeo, B. T. T., Smallwood, J., & Bernhardt, B. C. (2021). Differences in subcortico-cortical interactions identified from connectome and microcircuit models in autism. *Nature Communications*, *12*(1), Article 1. <https://doi.org/10.1038/s41467-021-21732-0>

Parsons, S., & McCormick, E. M. (2024). Limitations of two time point data for understanding individual differences in longitudinal modeling—What can difference reveal about change? *Developmental Cognitive Neuroscience*, *66*, 101353. <https://doi.org/10.1016/j.dcn.2024.101353>

Paus, T. (2010). Growth of white matter in the adolescent brain: Myelin or axon? *Brain and Cognition*, *72*(1), 26–35. <https://doi.org/10.1016/j.bandc.2009.06.002>

Paus, T., Keshavan, M., & Giedd, J. N. (2008). Why do many psychiatric disorders emerge during adolescence? *Nature Reviews Neuroscience*, *9*(12), Article 12. <https://doi.org/10.1038/nrn2513>

Quach, N. E., Yang, K., Chen, R., Tu, J., Xu, M., Tu, X. M., & Zhang, X. (2022). Post-hoc power analysis: A conceptually valid approach for power based on observed study data. *General Psychiatry*, *35*(4), e100764. <https://doi.org/10.1136/gpsych-2022-100764>

Raznahan, A., Lerch, P., Lee, N., Greenstein, D., Wallace, L., Stockman, M., Clasen, L., Shaw, W., & Giedd, N. (2011). Patterns of Coordinated Anatomical Change in Human Cortical Development: A Longitudinal Neuroimaging Study of Maturational Coupling. *Neuron*, *72*(5), 873–884. <https://doi.org/10.1016/j.neuron.2011.09.028>

Reiter, A. M. F., Moutoussis, M., Vanes, L., Kievit, R., Bullmore, E. T., Goodyer, I. M., Fonagy, P., Jones, P. B., & Dolan, R. J. (2021). Preference uncertainty accounts for developmental effects on susceptibility to peer influence in adolescence. *Nature Communications*, *12*(1), Article 1. <https://doi.org/10.1038/s41467-021-23671-2>

Sapouna, M., & Wolke, D. (2013). Resilience to bullying victimization: The role of individual, family and peer characteristics. *Child Abuse & Neglect*, *37*(11), 997–1006. <https://doi.org/10.1016/j.chiabu.2013.05.009>

Schmierer, K., Tozer, D. J., Scaravilli, F., Altmann, D. R., Barker, G. J., Tofts, P. S., &

Miller, D. H. (2007). Quantitative magnetization transfer imaging in postmortem multiple sclerosis brain. *Journal of Magnetic Resonance Imaging*, 26(1), 41–51. <https://doi.org/10.1002/jmri.20984>

Stüber, C., Morawski, M., Schäfer, A., Labadie, C., Wähnert, M., Leuze, C., Streicher, M., Barapatre, N., Reimann, K., Geyer, S., Spemann, D., & Turner, R. (2014). Myelin and iron concentration in the human brain: A quantitative study of MRI contrast. *NeuroImage*, 93, 95–106. <https://doi.org/10.1016/j.neuroimage.2014.02.026>

Sydnor, V. J., Larsen, B., Bassett, D. S., Alexander-Bloch, A., Fair, D. A., Liston, C., Mackey, A. P., Milham, M. P., Pines, A., Roalf, D. R., Seidlitz, J., Xu, T., Raznahan, A., & Satterthwaite, T. D. (2021a). Neurodevelopment of the association cortices: Patterns, mechanisms, and implications for psychopathology. *Neuron*, 109(18), 2820–2846. <https://doi.org/10.1016/j.neuron.2021.06.016>

Sydnor, V. J., Larsen, B., Bassett, D. S., Alexander-Bloch, A., Fair, D. A., Liston, C., Mackey, A. P., Milham, M. P., Pines, A., Roalf, D. R., Seidlitz, J., Xu, T., Raznahan, A., & Satterthwaite, T. D. (2021b). Neurodevelopment of the association cortices: Patterns, mechanisms, and implications for psychopathology. *Neuron*, 109(18), Article 18. <https://doi.org/10.1016/j.neuron.2021.06.016>

Teffer, K., & Semendeferi, K. (2012). Human prefrontal cortex. In *Progress in Brain Research* (Bd. 195, S. 191–218). Elsevier. <https://doi.org/10.1016/B978-0-444-53860-4.00009-X>

Van Harmelen, A.-L., Kievit, R. A., Ioannidis, K., Neufeld, S., Jones, P. B., Bullmore, E., Dolan, R., The NSPN Consortium, Fonagy, P., & Goodyer, I. (2017). Adolescent friendships predict later resilient functioning across psychosocial domains in a healthy community cohort. *Psychological Medicine*, 47(13), 2312–2322. <https://doi.org/10.1017/S0033291717000836>

Váša, F., Romero-Garcia, R., Kitzbichler, M. G., Seidlitz, J., Whitaker, K. J., Vaghi, M. M., Kundu, P., Patel, A. X., Fonagy, P., Dolan, R. J., Jones, P. B., Goodyer, I. M., the NSPN Consortium, Vértes, P. E., & Bullmore, E. T. (2020). Conservative and disruptive modes of adolescent change in human brain functional connectivity. *Proceedings of the National Academy of Sciences*, 117(6), 3248–3253. <https://doi.org/10.1073/pnas.1906144117>

Veer, I. M., Riepenhausen, A., Zerban, M., Wackerhagen, C., Puhmann, L. M. C., Engen, H., Köber, G., Bögemann, S. A., Weermeijer, J., Uściłko, A., Mor, N., Marciniak, M. A., Askelund, A. D., Al-Kamel, A., Ayash, S., Barsuola, G., Bartkute-Norkuniene, V., Battaglia, S., Bobko, Y., ... Kalisch, R. (2021). Psycho-social factors associated with mental resilience in the Corona lockdown. *Translational Psychiatry*, 11(1), 1–11. <https://doi.org/10.1038/s41398-020-01150-4>

Wang, F., Ren, S.-Y., Chen, J.-F., Liu, K., Li, R.-X., Li, Z.-F., Hu, B., Niu, J.-Q., Xiao, L., Chan, J. R., & Mei, F. (2020). Myelin degeneration and diminished myelin renewal contribute to age-related deficits in memory. *Nature Neuroscience*, 23(4), Article 4. <https://doi.org/10.1038/s41593-020-0588-8>

Weiskopf, N., Edwards, L. J., Helms, G., Mohammadi, S., & Kirilina, E. (2021). Quantitative magnetic resonance imaging of brain anatomy and in vivo histology. *Nature Reviews Physics*, 3(8), 570–588. <https://doi.org/10.1038/s42254-021-00326-1>

Wenger, E., Polk, S. E., Kleemeyer, M. M., Weiskopf, N., Bodammer, N. C., Lindenberger, U., & Brandmaier, A. M. (2022). Reliability of quantitative multiparameter maps is high for magnetization transfer and proton density but attenuated for R1 and R2* in healthy young adults. *Human Brain Mapping*, 43(11), 3585–3603. <https://doi.org/10.1002/hbm.25870>

Whitaker, K. J., Vértes, P. E., Romero-Garcia, R., Váša, F., Moutoussis, M., Prabhu, G., Weiskopf, N., Callaghan, M. F., Wagstyl, K., Rittman, T., Tait, R., Ooi, C., Suckling, J., Inkster, B., Fonagy, P., Dolan, R. J., Jones, P. B., Goodyer, I. M., the NSPN Consortium, & Bullmore, E. T. (2016). Adolescence is associated with genomically patterned consolidation of the hubs of the human brain connectome. *Proceedings of the National Academy of*

Sciences, 113(32), 9105–9110. <https://doi.org/10.1073/pnas.1601745113>

Xin, W., & Chan, J. R. (2020). Myelin plasticity: Sculpting circuits in learning and memory. *Nature Reviews Neuroscience*, 21(12), Article 12. <https://doi.org/10.1038/s41583-020-00379-8>

Xin, W., Mironova, Y. A., Shen, H., Marino, R. A. M., Waisman, A., Lamers, W. H., Bergles, D. E., & Bonci, A. (2019). Oligodendrocytes Support Neuronal Glutamatergic Transmission via Expression of Glutamine Synthetase. *Cell Reports*, 27(8), 2262-2271.e5. <https://doi.org/10.1016/j.celrep.2019.04.094>

Yang, G. J., Murray, J. D., Wang, X.-J., Glahn, D. C., Pearlson, G. D., Repovs, G., Krystal, J. H., & Anticevic, A. (2016). Functional hierarchy underlies preferential connectivity disturbances in schizophrenia. *Proceedings of the National Academy of Sciences*, 113(2), E219–E228. <https://doi.org/10.1073/pnas.1508436113>

Zemmar, A., Chen, C.-C., Weinmann, O., Kast, B., Vajda, F., Bozeman, J., Isaad, N., Zuo, Y., & Schwab, M. E. (2018). Oligodendrocyte- and Neuron-Specific Nogo-A Restrict Dendritic Branching and Spine Density in the Adult Mouse Motor Cortex. *Cerebral Cortex*, 28(6), 2109–2117. <https://doi.org/10.1093/cercor/bhx116>

Zhao, T. C., Corrigan, N. M., Yarnykh, V. L., & Kuhl, P. K. (2022). Development of executive function-relevant skills is related to both neural structure and function in infants. *Developmental Science*, 25(6), e13323. <https://doi.org/10.1111/desc.13323>

Zheng, Y.-Q., Zhang, Y., Yau, Y., Zeighami, Y., Larcher, K., Misic, B., & Dagher, A. (2019). Local vulnerability and global connectivity jointly shape neurodegenerative disease propagation. *PLOS Biology*, 17(11), e3000495. <https://doi.org/10.1371/journal.pbio.3000495>

Ziegler, G., Hauser, T. U., Moutoussis, M., Bullmore, E. T., Goodyer, I. M., Fonagy, P., Jones, P. B., NSPN Consortium, Lindenberger, U., & Dolan, R. J. (2019). Compulsivity and impulsivity traits linked to attenuated developmental frontostriatal myelination trajectories. *Nature Neuroscience*, 22(6), 992–999. <https://doi.org/10.1038/s41593-019-0394-3>

REVIEWERS' COMMENTS

Reviewer #1 (Remarks to the Author):

The authors did an excellent and very thorough job addressing reviewer comments, considerably strengthening an already very strong manuscript.

Reviewer #2 (Remarks to the Author):

The authors did an outstanding job addressing the reviewer feedback and significantly improved an already impressive paper. Kudos on a fabulous piece of research!

Reviewer #3 (Remarks to the Author):

I enjoyed reading the revision of the manuscript. The authors have thoughtfully dealt with my concerns regarding power and reliability of the measures used, as well as more conceptual issues. I also found the inclusion of Figure 3A to clarify the process of calculating the MIPC very useful.

However, I felt like my question of specificity had not been really addressed through analysis, or reference to the wider literature, but a longer elucidation of the putative mechanisms. I think it's important to stress this is a correlation, and assess if this correlation also holds with related measures (or not).

I think the choice of MTsat should be briefly described in the methods. The authors point to the sensitivity of using MT to detect individual differences but have cited a post-mortem paper, I just wanted to bring to their attention work by my colleagues and I on using MTsat and R2* to identify changes in neurodevelopmental conditions (Krishnan et al., 2022, eLife; Cler et al., 2021, Brain).

RESPONSE TO REVIEWERS (NCOMMS-24-08809A)

We would like to thank the Editors and Reviewers again for their positive evaluations and constructive comments. Please find our responses to the remaining comments below.

REVIEWERS' COMMENTS

Reviewer #1 (Remarks to the Author):

The authors did an excellent and very thorough job addressing reviewer comments, considerably strengthening an already very strong manuscript.

Thank you very much for your positive evaluation and the constructive comments you provided in the last review round.

Reviewer #2 (Remarks to the Author):

The authors did an outstanding job addressing the reviewer feedback and significantly improved an already impressive paper. Kudos on a fabulous piece of research!

Thank you very much for your kind words and useful comments in the last round of reviews. We are very happy to hear that you enjoyed our manuscript.

Reviewer #3 (Remarks to the Author):

I enjoyed reading the revision of the manuscript. The authors have thoughtfully dealt with my concerns regarding power and reliability of the measures used, as well as more conceptual issues. I also found the inclusion of Figure 3A to clarify the process of calculating the MIPC very useful.

Thank you very much for your helpful feedback and positive evaluation of the current project.

However, I felt like my question of specificity had not been really addressed through analysis, or reference to the wider literature, but a longer elucidation of the putative mechanisms. I think it's important to stress this is a correlation, and assess if this correlation also holds with related measures (or not).

Thank you for this comment. We have now included a wider literature on the topic in the discussion. We agree that the association between changes in resilience capacity and changes in MTsat is a correlation that can be influenced by various factors. We carefully discuss other influencing factors (such as a supportive social network) in the Discussion section. We hope that we correctly understood 'related measures' to refer to potential cognitive changes or changes in exposure to adversity that were requested in the last review round. From a conceptual perspective, we do not expect the observed association, or correlation, to be so specific to resilience capacities that it would be completely independent from cognitive changes. Rather, we expect the two to be strongly intertwined, such that cognitive maturation and the acquisition of new cognitive strategies facilitates resilience capacities. We follow your recommendation to include more literature on this topic in our Discussion section. We now include a very recent review on the neurobiology and systems biology of stress resilience (Kalisch et al., 2024), including a discussion on potential cognitive strategies, and address studies on 1. Prefrontal development and cognitive maturation (e.g., Nelson & Guyer, 2011; Teffer & Semendeferi, 2012), 2. Adaptive and maladaptive myelination in health and disease in humans and rodents (e.g., Chini & Hanganu-Opatz, 2021; Etkin et al., 2013; Knowles et al., 2022; Maas et al., 2020), 3. Effects of stress on the brain including the PFC (e.g., Kalisch et al., 2024; McEwen et al., 2016), 4. Effects of pathological prefrontal development in rodents and humans (e.g., Chini & Hanganu-Opatz, 2021), and 5. Prefrontal

neurobiological correlates of cognitive deficits in psychiatric disorders (e.g., Etkin et al., 2013; McTeague et al., 2017). While we cannot analytically address the potential role of cognitive strategies, we analyzed whether observed changes in MTsat could also be observed as a function of change in adversity exposure. We did not observe a significant relationship between change in the included adversity measures and change in MTsat. We now include this information in the Supplementary Material

We further stress that the observed association is based on a correlation that does not imply causality and may be influenced by other factors:

'The exact mechanisms underlying a potential protective effect cannot clearly be elucidated in the current study due to its correlational nature but may involve increased structural stability to stress-induced remodeling and enhanced cognitive maturation. Likely, resilient psychosocial development is closely coupled with the attainment of cognitive strategies that facilitate resilient outcomes (Kjelkenes et al., 2023). At the same time, environmental resilience factors such as social support facilitate resilient outcomes (McLaughlin et al., 2020; Reiter et al., 2021; Van Harmelen et al., 2017), and may in part exert their protective effect through an impact on brain maturational trajectories. Resilient outcomes are assumed to rely on a multi-modal and multi-faceted construct, acknowledging the environments we live in, but also other psychological variables beyond clinical symptomatology (such as positive affect, life satisfaction, personality traits). We cannot clearly disentangle the interaction of different intrinsic and extrinsic influences contributing to an individual's psychological well-being beyond resilience.'

I think the choice of MTsat should be briefly described in the methods. The authors point to the sensitivity of using MT to detect individual differences but have cited a post-mortem paper, I just wanted to bring to their attention work by my colleagues and I on using MTsat and R2* to identify changes in neurodevelopmental conditions (Krishnan et al., 2022, eLife; Cler et al., 2021, Brain).

We are happy to address this briefly in the manuscript. In addition to highlighting the utility of MTsat in the introduction, we now highlight the use of MTsat as well as the availability of other MPM sequences in the Methods section, citing your work alongside other in-vivo examples:

"MPM further comprises a set of other contrasts, such as R2 sensitive to iron content, but also PD and R1, yielding complementary insights into different aspects of tissue micro-architecture in vivo (Cler et al., 2021; Draganski et al., 2011; Krishnan et al., 2022; Weiskopf et al., 2013). Here, we focused on MT, which is considered a particularly strong in vivo marker of myelin with a high spatial correspondence with myelin basic protein and other myelin-related molecules in the brain, as has been verified by several histological validation studies (Mancini et al., 2020; Odrobina et al., 2005; Paquola & Hong, 2023; Schmierer et al., 2007). MT has further been demonstrated to show high reliability (Wenger et al., 2022) suitable for the study of individual differences and brain-behavior associations in vivo (Krishnan et al., 2022; Paquola et al., 2019; Ziegler et al., 2019)."*